🔓 | **Open Peer Review** | Bacteriology | Research Article

# SanA is an inner membrane protein mediating *Salmonella* Typhimurium infection

**Adrianna Stypułkowska,**[1] **Rafał Kolenda,**[1] **Ewa Carolak,**[1] **Joanna Czajkowska,**[1] **Agata Dutkiewicz,**[1] **Wiktoria Waszczuk,**[1] **Wiktoria Bińczyk,**[2] **Teresa L. M. Thurston,**[3] **Krzysztof Grzymajło**[1]

**ABSTRACT** Bacterial membrane proteins like SanA are essential for environmental interactions, significantly affecting the physicochemical properties of the bacterial envelope and influencing *Salmonella*'s antibiotic resistance and infection traits. Previous research links *sanA* deletion to increased *Salmonella* invasiveness, though the mechanisms are poorly understood. This study explores SanA's role in *Salmonella* infection both *in vitro* and *in vivo*. It examines its expression pattern, subcellular localization, and connection with the genetic background responsible for the infection phenotype following *sanA* knockout. Through subcellular fractionation and Western blotting, SanA was found mainly in the inner membrane. Transcriptional fusion indicated that *sanA* expression is important during late exponential and early stationary growth phases and remains significant 24 h post-host cell entry. Invasion assays showed that *sanA* deletion in bacteria grown to early stationary phase increased invasiveness, partly due to higher *sicA* expression regulated by nutrient availability. *In vivo* results supported these findings, with the *sanA* mutant exhibiting enhanced colonization of mouse organs but being outcompeted by the wild type in competitive infection. This study provides new insights into the role of SanA in *Salmonella*'s response to environmental stress, including hostile environments, emphasizing the importance of inner membrane proteins in shaping bacterial fitness and pathogenicity.

**IMPORTANCE** *Salmonella* poses significant global health and economic challenges. Its successful infection depends on complex interactions between the bacteria and host cells, involving various proteins in the bacterial envelopes. One such protein, SanA, plays a role in bacterial interaction with the environment, affecting antibiotic resistance and infection capability. Previous studies revealed that removing the *sanA* gene increases *Salmonella*'s ability to enter the host cells, though the underlying mechanisms were unclear. This research investigates SanA's role during infections, discovering its primary location in the inner bacterial membrane and its heightened activity during specific growth phases and post-host cell entry. Removing *sanA* made the bacteria more invasive, likely due to the upregulation of genes aiding host cell infection, especially in nutrient-rich conditions. In mouse infection experiments, SanA-deficient bacteria colonized organs more effectively but were less competitive when wild-type and mutant bacteria coexisted. This indicates SanA's role in managing environmental stress, enhancing *Salmonella*'s infection and survival capabilities.

**KEYWORDS** SanA, inner membrane, invasion, *Salmonella*, SPI-1, pathogenicity, infection

**Peer Reviewer** Yosef Daniel Huberman, Instituto Nacional de Tecnologia Agropecuaria, Buenos Aires, Argentina

Address correspondence to Krzysztof Grzymajło, krzysztof.grzymajlo@upwr.edu.pl.

The authors declare no conflict of interest.

See the funding table on p. 17.

*S*almonella enterica is a bacterial pathogen responsible for food-borne diseases, causing considerable morbidity and mortality in humans and livestock (1). A crucial aspect of *Salmonella*'s pathogenicity lies in its ability to adhere to and invade host

cells (2). The bacterium employs various structures for these processes, ranging from monomeric proteins to complex molecular machines. Notably, the type III secretion system (T3SS) encoded by genes in *Salmonella* Pathogenicity Islands I (SPI-1) and II (SPI-2) allows the bacteria to invade and survive within both phagocytic and non-phagocytic cells by translocating effector proteins into the host cell, altering vesicular trafficking and cytoskeletal dynamics (3).

*Salmonella* evades intracellular immune responses and persists in hostile environments through a sophisticated cell envelope that offers protection, facilitates nutrient intake, and expels waste (4). The envelope consists of an outer membrane (OM) and an inner membrane (IM), each embedded with specialized proteins. The OM, with its tightly packed lipopolysaccharide (LPS), features outer membrane proteins (OMPs) that function as selective barriers and environmental contact platforms (4). OMPs, including porins and efflux pumps, balance nutrient uptake and toxin exclusion, ensuring cellular homeostasis and defense against threats like xenobiotics (5). The IM, although less exposed to the external environment, contains inner membrane proteins (IMPs) crucial for processes like ATP synthesis and nutrient translocation (4). The interaction between OMPs and IMPs enhances the bacterial cell's permeability barrier and resistance to xenobiotics and antimicrobial agents (6).

Gram-negative pathogens, including *Salmonella*, modify their membranes to improve resilience to environmental stress and successfully establish infections. These strategies involve decreasing porin expression in response to toxic agents (7) or modification of LPS to avoid immune recognition and enhance resilience to antimicrobial peptides by altering lipid A phosphates and acylation (8). Additionally, membrane attributes like charge, hydrophobicity, and permeability modulate bacterial resistance to external stresses and indirectly affect pathogenicity. It has been shown that increased bacterial cell hydrophobicity enhances phagocytosis efficiency, whereas hydrophilic bacteria resist ingestion by phagocytes (9). Since several bacterial virulence factors, including T3SS-1, flagella, and chemotactic receptors, are essential components of the bacterial envelope, the relationship between membrane permeability and virulence is crucial. Previous studies have shown that the expression of *hilD,* a principal regulator of SPI-1, not only increases membrane permeability but also makes *Salmonella* more susceptible to membrane stress (10). This highlights the significant fitness cost that pathogenic bacteria face to balance virulence and survival.

Our current research builds on previous findings that demonstrated the role of SanA in modulating the properties of the bacterial membrane, affecting its charge and hydrophobicity, which in turn influences antibiotic resistance and enhances the intracellular survival of *Salmonella* Typhimurium (11). Additionally, the previous work demonstrated that a 10-nucleotide deletion in the *sanA* coding sequence is associated with enhanced bacterial invasive capabilities (12). However, the underlying mechanism remains unidentified. Hence, this study aims to examine SanA's expression during infection, its subcellular localization, and the molecular background responsible for the infection phenotype following *sanA* knockout. By examining the role of *sanA* in both *in vitro* and *in vivo* infection models, we seek to provide new insights into its impact on the infection process.

## RESULTS

### SanA is located in the inner membrane

The subcellular localization of SanA has not been determined so far, and our knowledge about its location within bacterial compartments is based solely on bioinformatic prediction tools. Furthermore, the results of these analyses are not consistent, indicating that SanA can be an outer or an inner membrane protein (11). To explore this aspect, we aimed to analyze the subcellular localization of SanA in *Salmonella*.

SanA was identified solely in the inner membrane fraction, co-located in this compartment with the control protein LepB (Fig. 1). The absence of SanA expression in the Δ*sanA* deletion mutant confirmed the specificity of the newly raised SanA antibody,

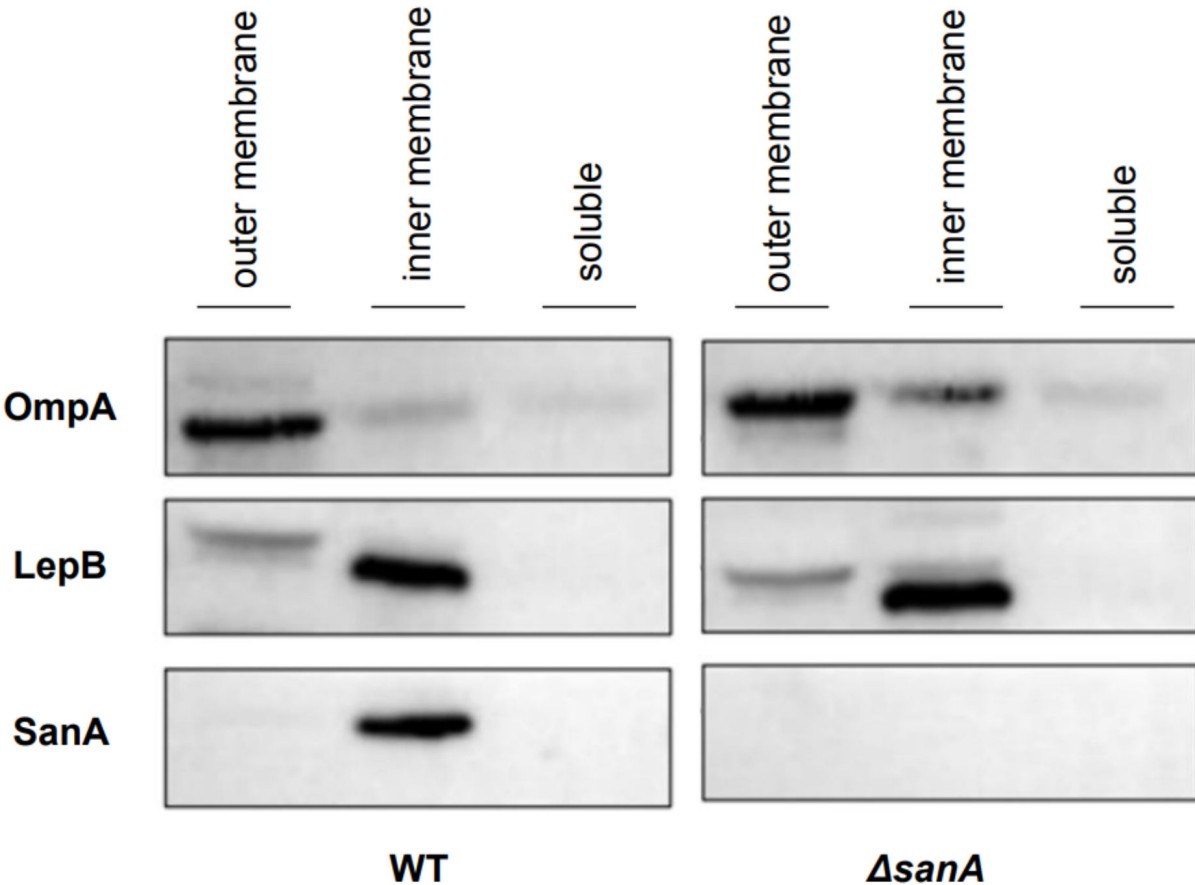

**FIG 1** SanA is situated within the inner membrane. Western blot analysis of wild-type (WT) and Δ*sanA S.* Typhimurium 4/74 fractions isolated using the lysozyme-EDTA method. After fractionation, proteins were subjected to SDS-PAGE, and membranes were probed with the indicated antisera. Proteins from the same number of cells were electrophoresed on each lane.

evidenced by the absence of non-specific binding. Furthermore, the membrane marker proteins LepB (inner membrane) and OmpA (outer membrane) were detected predominantly in their respective fractions (Fig. 1). This observation suggests that the cytoplasmic membrane integrity was maintained without significant disruption during the process of fractionation, ensuring the reliability of the experiment results.

## SanA expression is growth phase dependent and correlates with intracellular survival within macrophages

To examine the impact of diverse *in vitro* conditions on *sanA* expression, an analysis employing transcriptional fusion was carried out. The addition of the transcriptional *luc* reporter did not affect the growth kinetics of the bacteria (Fig. S1). We observed strong induction of the reporter in bacteria entering the LEP (late exponential phase) and ESP (early stationary phase) (Fig. 2A). Only a low level of reporter activity was detected in bacteria during the LSP (late stationary phase) (Fig. 2A). Upon culturing bacteria in Mg-MES pH 5.0 medium, resembling conditions in infected macrophages, the activity level was comparable to that in the EEP (early exponential phase) (Fig. 2A).

Similarly, to determine whether *sanA* is significantly expressed at a specific stage following entry into host cells and to verify the optimal time point post-infection for conducting the functional assay, macrophages were infected with the reporter strain. Subsequently, cells were lysed at different intervals post-infection for the quantification of luciferase activity. The *sanA* expression was the lowest and comparable between 2 h and 8 h post-infection ($P = 0.9950$) (Fig. 2B), despite some degree of replication at 8 h

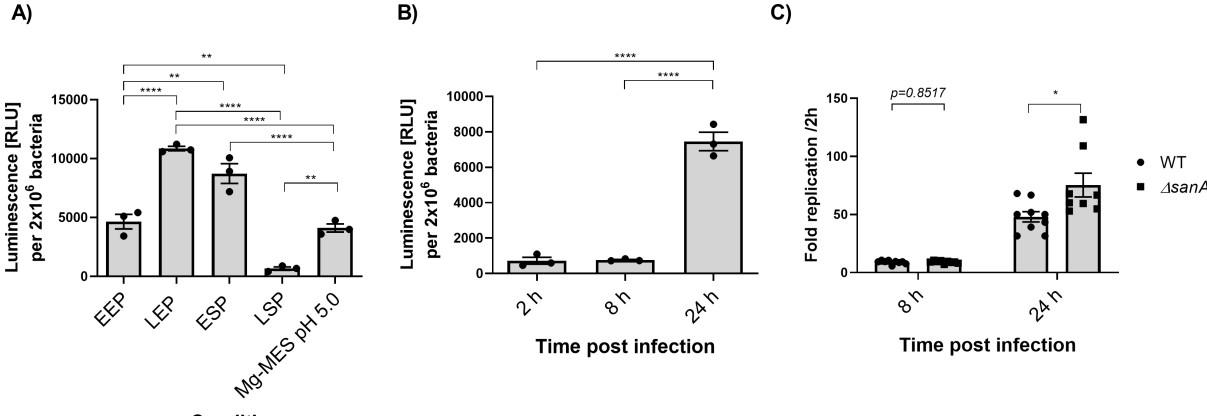

**FIG 2** The expression of SanA is dependent on the growth phase and is associated with intracellular survival within macrophages. (A) *In vitro* expression of a *luc* reporter fused to the *sanA* (*sanA*$_{RBS}$::*luc*) in different growth phases; EEP (OD$_{600}$ = 0.5); LEP (OD$_{600}$ = 1.0); ESP (OD$_{600}$ = 2.0); LSP (16 h culture); Mg-MES pH 5.0—SPI-2-inducing conditions, mimicking the environment of infected macrophages. (B) *In vitro* expression of a *luc* reporter fused to the *sanA* (*sanA*$_{RBS}$::*luc*) during the infection of immortalized bone marrow-derived murine macrophage (BMDM) isolated from C57BL/6 mice. (C) Intracellular survival within immortalized BMDM isolated from C57BL/6 mice of *S*. Typhimurium 4/74 and *sanA* mutant. The data are shown as mean values and SEM of three separate experiments. Statistical differences were analyzed by one-way analysis of variance with Tukey's correction (*, $P < 0.05$; **, $P < 0.01$; ***, $P < 0.001$; ****, $P < 0.0001$).

(Fig. 2C). The highest expression was detected after 24 h ($P < 0.0001$) (Fig. 2B), at which point bacteria had undergone further replication within macrophages (Fig. 2C). At this 24 h time point, *sanA* mutant bacteria showed significantly elevated intramacrophage survival compared to wild-type (WT) bacteria ($P = 0.0340$) (Fig. 2C).

## SanA deletion increases the invasion of intestinal epithelial cells and macrophages

Given prior studies indicating a possible involvement of SanA in the initial stages of infection, we utilized invasion assays to examine this phenomenon (12). Human epithelial cell line Caco-2 and immortalized bone marrow-derived murine macrophages (iBMDM) were infected with SPI-1-induced *Salmonella* strains at a multiplicity of infection (MOI) = 100 and MOI = 10, respectively. We noticed that the number of invading WT bacteria was significantly lower than Δ*sanA*, which revealed more than 42% and about 16% higher invasiveness toward Caco-2 and iBMDM, respectively ($P = 0.0035$; $P = 0.0049$) (Fig. 3A and B). Moreover, the complementation of mutation *in trans* restored the WT phenotype. To ensure that the observed effects were specific to *sanA* complementation and not due to the plasmid backbone, we utilized an empty plasmid as a control showing that Δ*sanA*-pWSK29 invaded both cell types significantly better than Δ*sanA*-pWSK29-*sanA* ($P = 0.0440$; $P = 0.0009$) (Fig. 3A and B).

## SanA deletion correlates with enhanced expression of *sicA*

As we observed higher invasiveness of *Salmonella* in a *sanA* knockout, we decided to analyze the molecular basis responsible for this phenotype. Keeping in mind that SPI-1 is the main determinant responsible for the invasion of non-phagocytic host cells, our examination included a comparative analysis of type III secretion-associated chaperone SicA promoter activity across populations and SicA expression in the WT and Δ*sanA*. Given SicA's crucial role and the fact that its expression is co-regulated with the components of SPI-1 to ensure the coordinated regulation of *Salmonella*'s invasion machinery, it is believed that *sicA* expression correlates with the expression of the entire SPI-1 system (13).

For SicA, the differences were detected in the ESP, whereas very low expression with no differences between analyzed strains was detected in the EEP (Fig. 4A). These results were examined quantitatively in densitometry analysis, where the average relative density of SicA was assessed against the relative density of green fluorescent protein

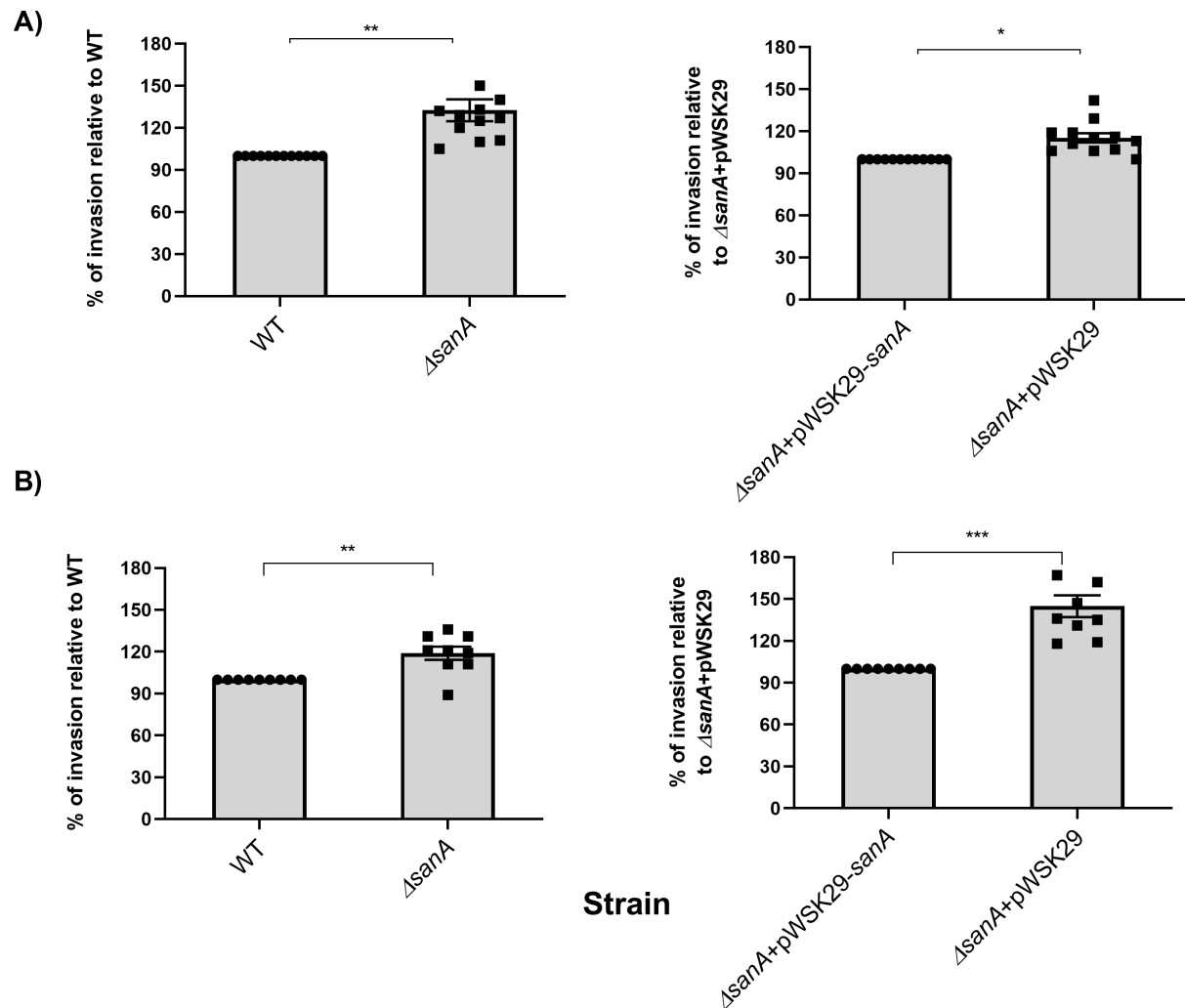

**FIG 3** SanA deletion increases the invasion of intestinal epithelial cells and macrophages. Invasion assay of (A) Caco-2 cell line and (B) immortalized bone marrow-derived macrophages isolated from C57BL/6 mice infected by WT and *sanA* mutant *S.* Typhimurium 4/74, and Δ*sanA* transformed with empty pWSK29 plasmid or vector with *sanA*. The data are shown as mean values and SEM of three separate experiments with statistical differences determined by Student's *t*-test (*, $P < 0.05$; **, $P < 0.01$; ***, $P < 0.001$).

(GFP), used as a protein load control in the bacterial lysates (Fig. S2). These findings are consistent with our flow cytometry analysis of the reporter system based on the GFP signal under the control of the promoter of interest and constitutive expression of mCherry. This examination showed a greater proportion of the Δ*sanA* population expressed *sicA*, particularly noticeable during the ESP ($P < 0.0001$) (Fig. 4B; Fig. S3). In this phase, approximately 72% of the WT population and about 95% of the Δ*sanA* population expressed *sicA* (Fig. 4B; Fig. S3).

## SicA expression in Δ*sanA* is regulated in a nutrient-dependent manner

As high levels of nutrients, resulting from improved accessibility or enhanced transport to bacteria, are associated with heightened virulence in pathogens (14, 15), we tested whether nutrient availability might impact *sicA* expression in WT and *sanA* mutant bacteria. To address this hypothesis, we employed our *sicA*p-GFP reporter system. When we grew cells in the conditions corresponding to those used in the infection assay (lysogeny broth [LB] containing 0.5% yeast extract), the *sicA* promoter was active in 83% of the Δ*sanA* population and only in 62% of the WT population ($P = 0.0037$) (Fig. 5; Fig. S4). In the absence of yeast extract, the activity of the *sicA* promoter was comparable

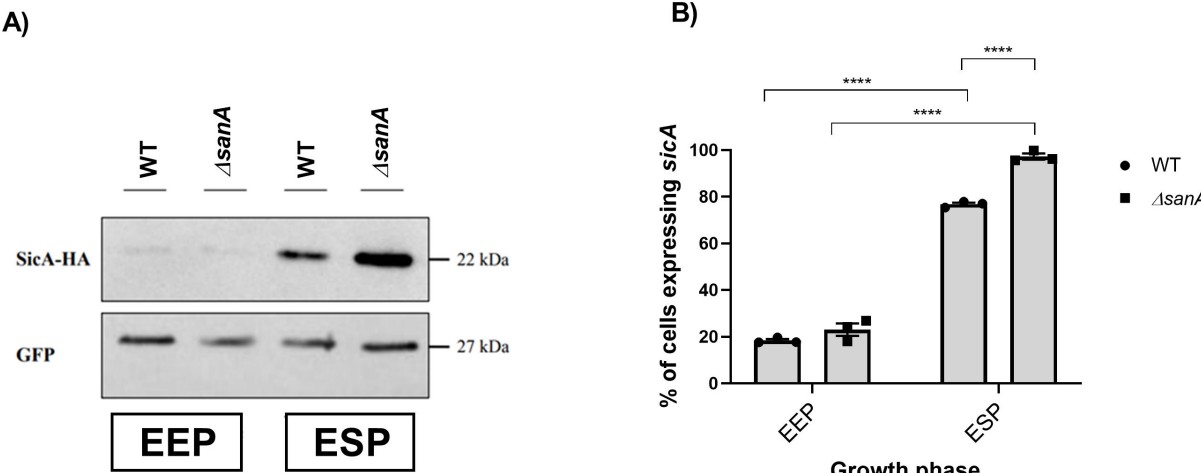

**FIG 4** SicA expression is elevated in the absence of SanA. (A) Determination of SicA expression by Western blotting at EEP corresponding to $OD_{600}$ = 0.5 and ESP corresponding to $OD_{600}$ = 2.0 in WT and *sanA* mutant 4/74 *S. Typhimurium*. (B) Fraction of cells expressing *sicA*-promoter dependent GFP ("ON" state) during growth in lysogeny broth medium until EEP or ESP. The fraction of cells in the ON state was determined relative to the negative control (100% in the "OFF" state), which consisted of the measured fluorescence of cells not expressing GFP. The data are shown as mean values and SEM of three separate experiments. Statistical differences were analyzed by two-way analysis of variance with Tukey's correction (****, $P < 0.0001$).

between both strains and reached less than 66% of the populations in the "ON" state (Fig. 5; Fig. S4). This activity level appeared to represent a threshold that was maintained independently of nutrient availability (Fig. 5; Fig. S4). In LB containing 2% yeast extract, we observed the highest (>86%) population of cells where the *sicA* promoter was active for both WT and Δ*sanA*.

## SanA deletion enhances colonization of the lymph glands and liver in BALB/c mice

Based on our *in vitro* findings demonstrating SanA's role in *Salmonella* invasion through enhanced *sicA* expression, we employed SPI-1-inducing bacteria to assess colonization in mice *in vivo*. Five days post-infection, we noted significantly lower bacterial loads in the liver and lymph glands of animals infected with WT compared to Δ*sanA* (Fig. 6A and B). The liver of WT-infected animals harbored approximately $10^4$ colony forming units (CFUs) of *Salmonella* per gram, while the Δ*sanA* group exhibited around $10^5$ CFUs, indicating a roughly 10-fold reduction in colonization ($P = 0.0186$) (Fig. 6A). Moreover, three mice in the WT group and only one in the Δ*sanA* group had no detectable bacterial presence (Fig. 6A). The Δ*sanA* strain also colonized the lymph glands significantly more effectively than the WT strain ($P = 0.0014$), with no detectable colonies in the WT group and over $10^4$ CFUs per gram in the Δ*sanA*-infected tissue (Fig. 6B). The highest bacterial counts were observed in the spleen, with approximately $10^5$ CFU/g for the WT and $10^7$ CFU/g for the Δ*sanA*, while no bacteria were detected in two mice from each group (Fig. 6C). Although the difference in colonization was not statistically significant ($P = 0.0945$), a trend toward increased colonization by the Δ*sanA* strain was observed (Fig. 6C). Colonization of the small intestine was statistically comparable between the two strains, with approximately $5 \times 10^3$ CFU/g for the WT and $10^4$ CFU/g for the Δ*sanA* ($P = 0.4771$). However, five mice from the WT group showed no bacteria in this tissue compared to only two from the Δ*sanA* group (Fig. 6D).

## SanA deletion decreases the virulence and fitness of *S.* Typhimurium in BALB/c mice

The virulence and fitness of WT and Δ*sanA* strains *in vivo* were investigated using a BALB/c infection model with intragastric administration of *Salmonella*. Bioluminescent

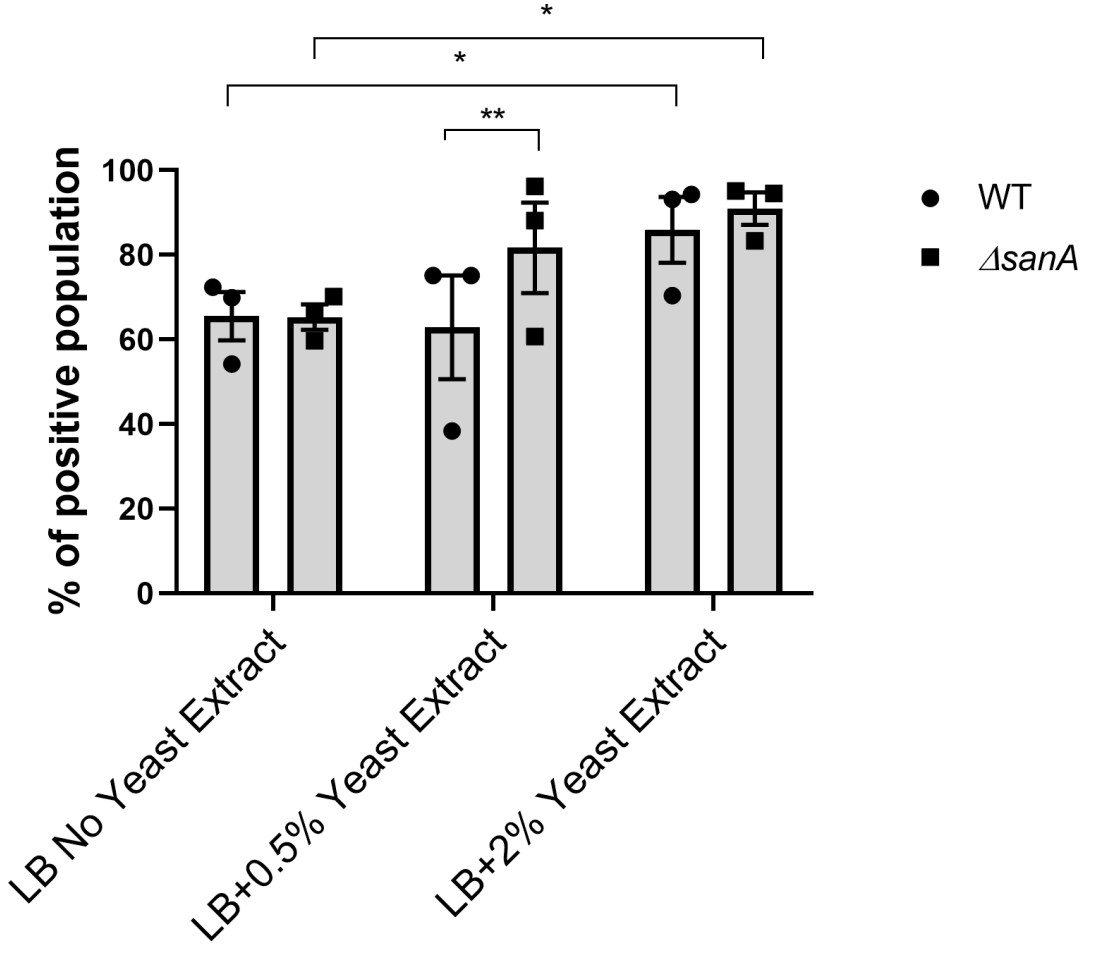

**FIG 5** Excess nutrients drive *sicA* expression. Fraction of cells expressing *sicA* ("ON" state) during growth in LB medium with different concentrations of yeast extract. The fraction of cells in the ON state was determined relative to the negative control (100% in the "OFF" state), which consisted of the measured fluorescence of cells not expressing the GFP (without *sicA* promoter). The data are shown as mean values and SEM of three separate experiments. Statistical differences were analyzed by two-way analysis of variance with Tukey's correction (*, $P < 0.05$; **, $P < 0.01$).

imaging (BLI) technology revealed significant differences in virulence patterns between these two strains. Mice infected with the Δ*sanA*::p16Slux exhibited prolonged infection-free periods compared to those inoculated with the WT::p16Slux ($P = 0.0036$) (Fig. 7A). Oral administration of $10^7$ CFU of the WT::p16Slux strain resulted in 100% infection (seven of seven mice) by day 10. In contrast, 71% (5 of 7) of mice infected with the Δ*sanA*::p16Slux strain showed bioluminescence signals 4 days later, with 29% (two of seven) remaining free of *Salmonella* by day 21 (Fig. 7A and B). The presence of *Salmonella*, as detected through BLI, was verified by plating homogenates from the spleen, liver, lymph glands, and small intestines. It confirmed that bioluminescent signals were indicative of *Salmonella* in these tissues (Fig. S5).

## Deletion mutant Δ*sanA* is outcompeted by *S.* Typhimurium WT in BALB/c mice

To directly compare the competitive index (CI) of the WT and Δ*sanA*, we assessed the number of viable bacteria recovered from the liver, spleen, small intestine, and lymph glands of mice 5 days post-infection (Fig. 8). Our CI analysis demonstrated that the Δ*sanA* mutant had significantly lower colonization efficiency in the liver and spleen compared

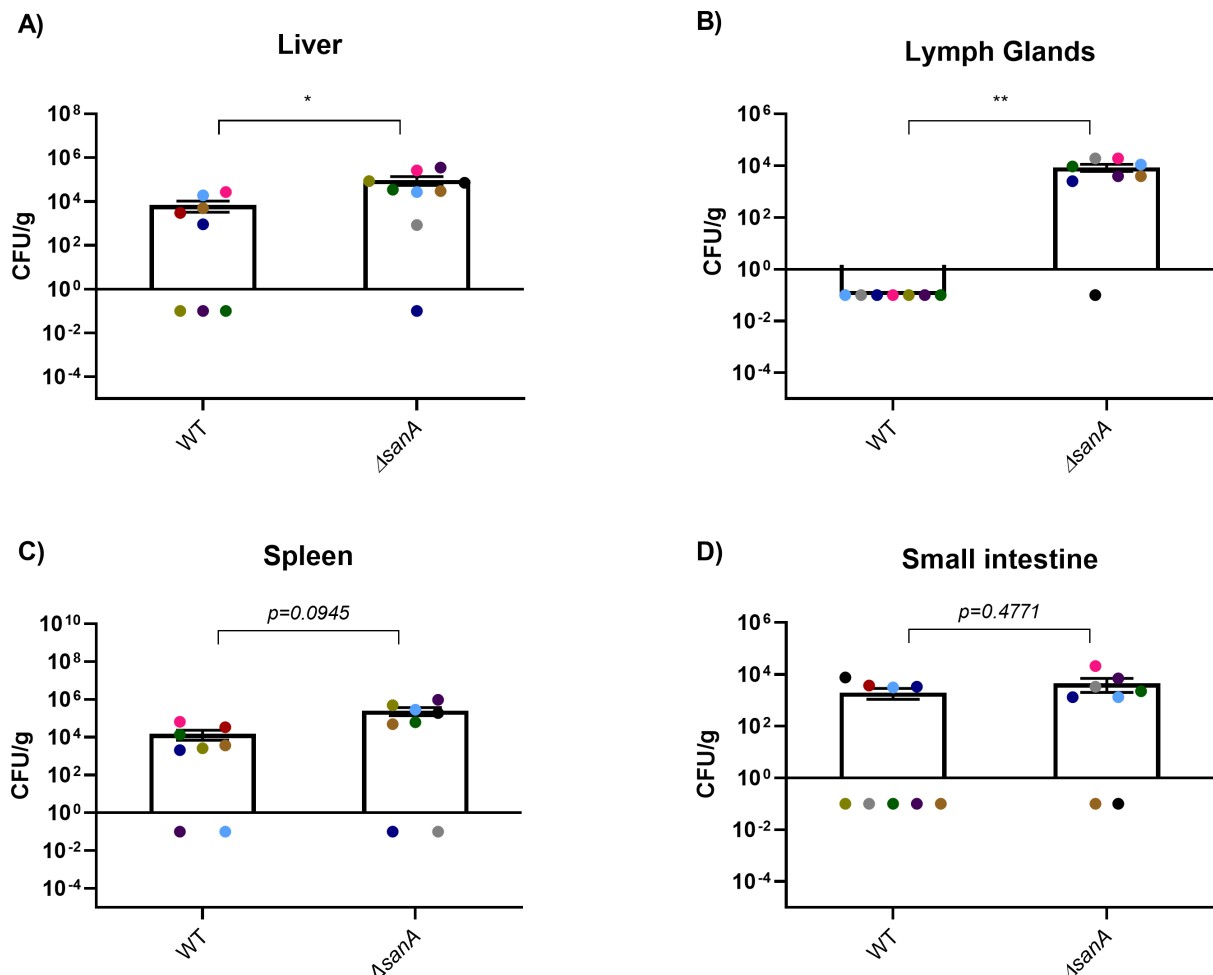

**FIG 6** The removal of SanA leads to increased colonization of lymph glands and liver in BALB/c mice. Colonization of mouse organs by WT and *sanA* mutant *S.* Typhimurium 4/74: (A) liver, (B) lymph glands, (C) spleen, and (D) small intestine. BALB/c mice (10 per group) were inoculated with $10^7$ CFU of *Salmonella* and sacrificed 5 days post-infection. Each colored dot represents an individual mouse, and the results are presented as mean values with SEM error bars. Since the scale has been logarithmically transformed, mice with no detected colonization were assigned a value of 0.1. Statistical differences were analyzed using the Mann-Whitney U-test after the removal of outliers (ROUT, Q = 1%) (*, $P < 0.05$; **, $P < 0.01$).

to the WT, with an average of 25 times less CFU/g ($P = 0.0292$; $P = 0.0261$). Although the Δ*sanA* also exhibited a trend toward reduced colonization in the small intestine and lymph glands, these differences were not statistically significant ($P = 0.1543$; $P = 0.5148$) (Fig. 8A).

Following streptomycin (Strep) treatment, the liver and spleen were more effectively colonized by the WT strain compared to the Δ*sanA* ($P = 0.0292$; $P = 0.0261$) (Fig. 8B and C). The lowest bacterial counts were observed in the lymph glands, with $10^6$ CFU/g for WT and $10^5$ CFU/g for Δ*sanA*, whereas three mice from the WT group and four from the Δ*sanA* group showed no bacteria (Fig. 8D). In turn, the small intestine was the most effectively colonized organ, with approximately $10^{10}$ CFU/g of WT and $10^9$ CFU/g of Δ*sanA* (Fig. 8E). While we examined bacterial counts in feces, we observed that the number of Δ*sanA* decreased by approximately 10-fold 3 days post-infection compared to the 1st day (Fig. 8F and G). In contrast, the WT bacterial count remained consistent throughout the experiment and was significantly higher than the Δ*sanA* count ($P = 0.0147$ after the 1st day and $P = 0.0335$ after the 3rd day) (Fig. 8F and G).

## A)

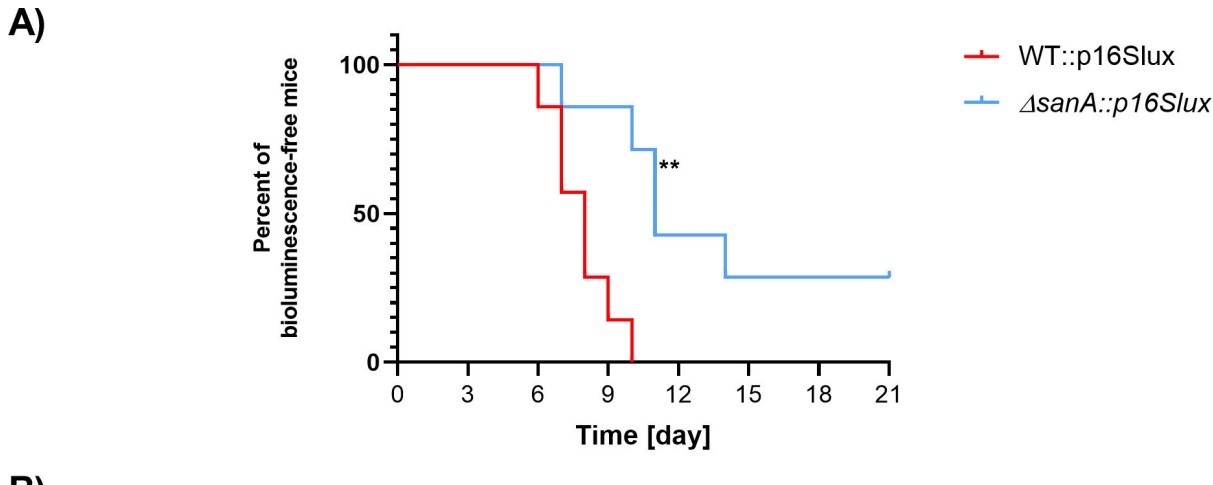

## B)

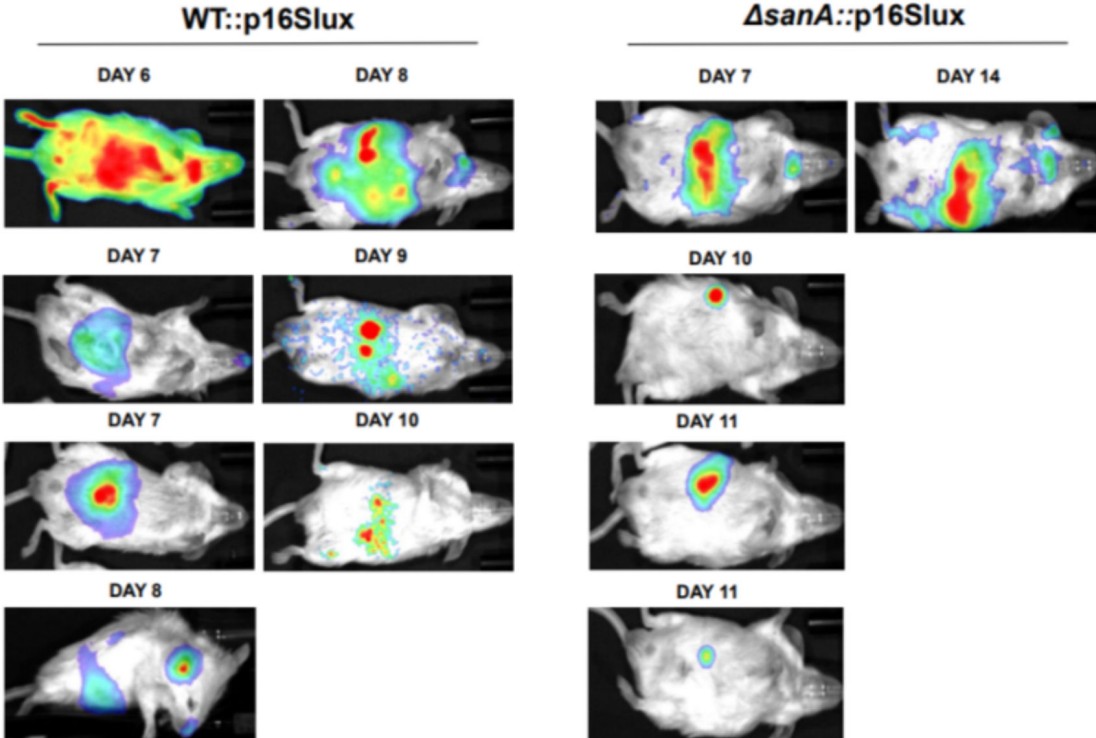

**FIG 7** SanA deletion reduces the virulence and fitness of *S.* Typhimurium in BALB/c mice. (A) Kaplan-Meier bioluminescence-free time curves for mice administered with WT::p16Slux and Δ*sanA*::p16Slux *S.* Typhimurium 4/74. The log-rank test (Mantel-Cox) was used for statistical analysis of the bioluminescence-free time curves (**, $P < 0.01$). (B) Monitoring of WT::p16Slux and Δ*sanA*::p16Slux *S.* Typhimurium 4/74 infections in mice using whole-body BLI. BALB/c mice (seven per group) were inoculated with $10^7$ CFU of *Salmonella*. The presence of bioluminescence signals was monitored in the animals every morning for 21 days post-inoculation. The intensity of bioluminescence emission is represented as a pseudocolor image (red indicating the highest intensity and blue the lowest).

## DISCUSSION

Bacterial membranes are composed of numerous proteins crucial in the interaction between the pathogen, the environment, and the host (16, 17). Among these molecules, SanA was first described by Rida et al. in 1996, who identified it as a protein contributing to vancomycin resistance. Specifically, they noted that overexpression of *sanA* reduced the vancomycin sensitivity of *Escherichia coli* mutant with an unidentified envelope

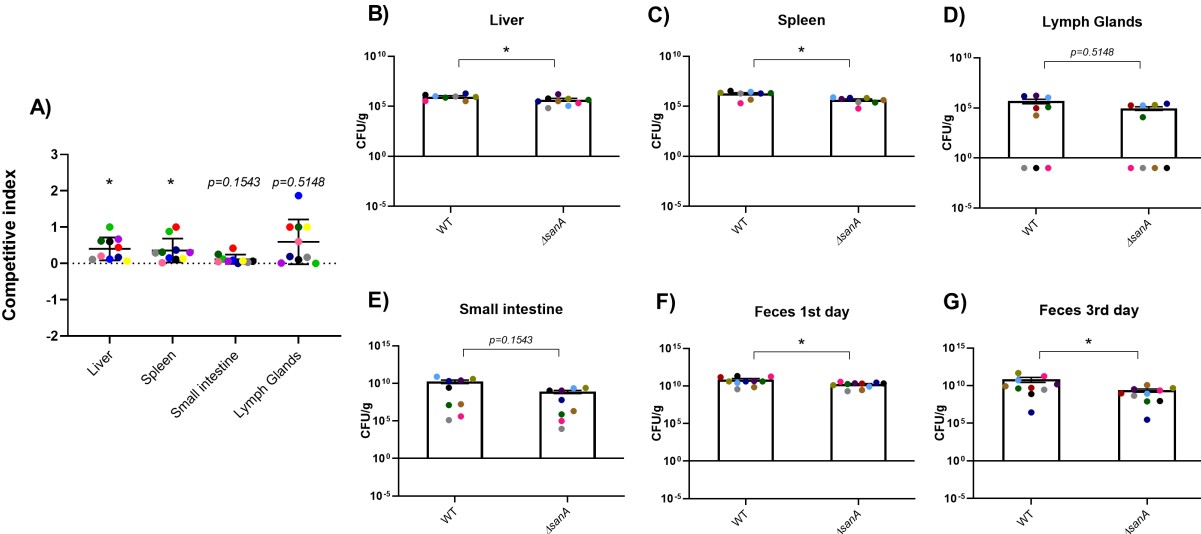

**FIG 8** Deletion mutant Δ*sanA* is outcompeted by WT in BALB/c mice. Bacterial counts and CI in respective organs. (A) CI was evaluated by quantifying viable bacteria recovered from these organs 5 days post-infection. Specifically, CI was calculated as the ratio of CFUs of Δ*sanA::kanR* (counted from kanamycin agar plates) to CFUs of the WT. A CI value of 1 indicates equal virulence. CFU/g was determined in (B) liver, (C) spleen, (D) lymph glands, (E) small intestine, (F) feces 1st day post-infection, and (G) feces 3rd day post-infection. Each data point represents an individual mouse, and the results are presented as mean values with SEM error bars. Since the scale has been logarithmically transformed, mice with no detected colonization were assigned a value of 0.1. Statistical differences were analyzed using the Mann-Whitney U-test after the removal of outliers (ROUT, Q = 1%) (*, $P < 0.05$).

permeability defect (18). Similarly, *S*. Typhimurium's SfiX, an ortholog of SanA, was found to mitigate the cell division defect caused by HisHF overproduction (19).

In our previous studies, we identified a role for SanA in antibiotic resistance, suggesting the association of this phenotype with the physicochemical properties of the bacterial membrane (11). Furthermore, we showed that a 10-nucleotide mutation in the *sanA*-encoding sequence results in an enhanced invasion of intestinal epithelial cells of human origin (12). However, the mechanisms underlying these phenotypes remain unexplored. Therefore, we aimed to characterize the impact of SanA on *S*. Typhimurium 4/74 infection and elucidate its role in membrane dynamics and pathogenicity.

Although SanA affects the physicochemical properties of the bacterial membrane, its subcellular localization remains unverified (11). Here, we demonstrate for the first time that SanA is an inner membrane protein. This suggests that SanA can be primarily involved in the interactions with the outer membrane components rather than the direct maintaining of inner membrane integrity. This situation may be parallel to TolA, which is an inner membrane protein affecting the membrane integrity through its carboxyl-terminal domain (20, 21). In line with our observations, mutation in the *tolA* has also been reported to result in increased outer membrane permeability (21).

To explore SanA's role in infection, we utilized three models: human intestinal epithelial cells, immortalized mouse macrophages, and a murine infection model. Under SPI-1-inducing conditions, *sanA* deletion significantly increased *Salmonella* invasion of epithelial cells and macrophages (22). Consistent with *in vitro* results, *in vivo* data revealed that Δ*sanA* showed increased colonization of the liver and lymph glands, suggesting enhanced proliferation within the host. This phenotype corresponds with our previous observations of heightened invasiveness linked to a 10-nucleotide *sanA* mutation (12), reflects a link between *sanA* expression and infection dynamics, and suggests an association with SPI-1, which is a crucial factor responsible for *Salmonella* invasion (23, 24). We observed that under SPI-1-inducing conditions, *sanA* expression peaked, mirroring the RNA-level findings of Kröger et al. (25), indicating its role during the early stages of infection (25). Conversely, under SPI-2 conditions, characterized by

low SanA levels, no differences in invasiveness were observed, further supporting an SPI-1-specific impact (11).

Our investigation into the regulatory dynamics of virulence genes revealed that ΔsanA exhibited significantly higher expression of sicA, a key SPI-1 component, under conditions that mimic the early stages of infection. This observation was validated using a reporter assay, which showed a marked increase in sicA expression in the ΔsanA strain compared to WT. The upregulation of sicA correlated with increased invasion, reflecting a strong link between membrane dynamics and virulence gene regulation (13, 22, 26). This expression pattern suggests that SanA plays a role in maintaining membrane integrity during high virulence gene expression, potentially preventing the destabilizing effects of hyperactive T3SS machinery (23).

In macrophages, ΔsanA strains demonstrated increased intracellular replication 24 h after infection, a pattern consistent with our previous studies in bone marrow-derived macrophages (11). This suggests that SanA may indirectly modulate nutrient availability, enhancing survival in the nutrient-limited Salmonella-containing vacuole (27). A more permeable membrane facilitates more efficient nutrient transport, supporting bacterial growth in these restrictive environments. Additionally, we noted that despite higher organ loads, mice infected with the ΔsanA strain had elevated bioluminescence-free time rates compared to WT-infected mice. This supports the hypothesis that increased permeability heightens immune recognition, facilitating an immune response that curtails bacterial dissemination without causing lethal infection (28, 29).

Recent research unveiled intriguing connections between nutrient availability and gene expression in bacteria. For example, it has been observed that nutrients such as yeast extract can induce the expression of the SPI-1 gene in Salmonella (15, 24). Our findings show that sicA expression in ΔsanA was markedly higher in the presence of 0.5% yeast extract, a condition similar to that used in infection assays. This suggests that enhanced permeability in ΔsanA facilitates increased nutrient uptake, triggering sicA induction. Notably, under high nutrient conditions (2% yeast extract), sicA expression showed no significant differences between the strains. We propose it may be attributed to the abundant nutrient availability supporting efficient transport across the membrane in both strains, despite the enhanced membrane integrity of the WT. Similarly, in the absence of yeast extract, sicA activity remained comparable between strains, suggesting the presence of a threshold level that is maintained regardless of nutrient availability. These results indicate that sanA deletion influences virulence gene regulation in a nutrient-dependent manner, potentially linking membrane permeability to environmental sensing and adaptation (15, 30).

We demonstrated earlier that the deletion of sanA leads to increased membrane permeability, suggesting it may drive higher expression of SPI-1 genes due to compensatory stress responses, as seen in other studies linking envelope stress to virulence gene regulation (11, 31). This heightened SPI-1 expression, however, imposes a fitness cost, as observed in CI experiments where WT strains outcompeted ΔsanA in the mice infection model. The mutant's increased initial colonization appears to come at the expense of long-term fitness, possibly due to resource misallocation under competitive conditions (32).

Overall, our study highlights SanA's role in Salmonella's response to stress and virulence gene expression. Its role in modulating membrane permeability directly affects pathogenicity by balancing nutrient transport and membrane integrity, influencing sicA expression, and shaping infection outcomes. Understanding this interplay offers new insights into the role of inner membrane proteins in bacterial adaptation and pathogenesis, paving the way for further research in this field. It should investigate SanA's role in host-cell trafficking using microscopy and trafficking markers, as well as its impact on intracellular replication. The increased bacterial loads in lymph glands and liver raise questions about whether SanA influences persistence, which can be explored through immune response analysis in infected mice. Finally, dissecting the regulatory pathways linking SanA to virulence gene expression will clarify how membrane integrity influences

pathogenicity, providing a deeper understanding of SanA's role in bacterial adaptation and infection.

## MATERIAL AND METHODS

### Bacteria, plasmids, and growth conditions

Tables 1 to 3 contain the complete list of bacterial strains, plasmids, and primers used in this study, respectively. All the *Salmonella* strains employed were derived from the *Salmonella enterica* serovar Typhimurium 4/74 WT. Unless specified otherwise, bacterial cultures were typically cultivated at 37°C for 16 h (overnight) either in LB under dynamic conditions with shaking (180 rpm) or on agar plates. For all invasion studies, *Salmonella* strains were grown under SPI-1-inducing conditions (ESP in LB medium with aeration; $OD_{600} = 2.0$) (33). When activation of SPI-2 was required, the bacterial strains were grown overnight in LB medium and then washed in Mg-MES minimal medium (consisting of 170 mM 2-(N-morpholino)ethanesulfonic acid [MES] at pH 5.0, 5 mM KCl, 7.5 mM $(NH_4)_2SO_4$, 0.5 mM $K_2SO_4$, 1 mM $KH_2PO_4$, 10 mM $MgCl_2$, 38 mM glycerol, and 0.1% casamino acids), with the pH adjusted to 5.0 (34). The bacteria were then grown for 6 h in the same medium.

If required, antibiotics were supplemented at specific concentrations: 100 µg/mL for ampicillin, 50 µg/mL for kanamycin, 500 µg/mL for erythromycin, and 50 µg/mL for Strep. For inducing *lac* and *ara* promoters, isopropylthio-β-galactoside was added to a final concentration of 0.5 mM or L-arabinose to a final concentration of 0.2%, respectively. Cell growth was analyzed using optical density readings at 600 nm.

### Cell culture

Human intestinal epithelial cell line Caco-2 (DMSZ, Germany) was grown at 37°C with 5% $CO_2$ in Dulbecco's Modified Eagle's Medium (DMEM)/Ham's F-12 supplemented with 1 mM L-glutamine, 100 U/mL penicillin-streptomycin, and 10% fetal bovine serum (FBS) and passaged according to standard ATCC protocols. For infection assays, cells were seeded in 24-well plates at a density of $1.2 \times 10^5$ cells and used in experiments once monolayers were established.

iBMDMs were maintained in DMEM high glucose supplemented with 20% (vol/vol) of L929-MCSF supernatant (L-cell conditioned medium [LCM]), 10% (vol/vol) of FBS, 10 mM of HEPES, 1 mM of sodium pyruvate, 0.05 mM of β-mercaptoethanol, and 100 U/mL of

**TABLE 1**  Bacterial strains used in this study

| Strain | Relevant feature(s) | Reference |
|---|---|---|
| *S.* Typhimurium 4/74 (WT) | Wild-type serovar Typhimurium (WT) | Dr. Derek Pickard, Cambridge Institute for Therapeutic Immunology & Infectious Disease, University of Cambridge Department of Medicine, Cambridge, UK |
| *S.* Typhimurium *sanA*<sub>RBS</sub>::*luc* | *sanA*<sub>RBS</sub>::*luc* | This study |
| *S.* Typhimurium::p16Slux | WT::p16Slux | This study |
| *S.* Typhimurium pFCcGi-p*sicA* | pFCcGi-p*sicA* | This study |
| *S.* Typhimurium pFPV25.1GFPmut3Kan-*sicA*-2xHA | pFPV25.1GFPmut3Kan-*sicA*-2xHA | This study |
| *S.* Typhimurium 4/74 Δ*sanA* | *S.* Typhimurium 4/74 with *sanA* gene knockout (Δ*sanA*) | (11) |
| *S. Typhimurium* Δ*sanA::kanR* | Δ*sanA::kanR* | (11) |
| *S.* Typhimurium Δ*sanA* pWSK29 | pWSK29 | (11) |
| *S.* Typhimurium Δ*sanA* pWSK29-*sanA* | pWSK29-*sanA* | (11) |
| *S.* Typhimurium Δ*sanA*::p16Slux | Δ*sanA*::p16Slux | This study |
| *S.* Typhimurium Δ*sanA* pFCcGi-p*sicA* | pFCcGi-p*sicA* | This study |
| *S.* Typhimurium Δ*sanA* pFPV25.1GFPmut3Kan-*sicA*-2xHA | pFPV25.1GFPmut3Kan-*sicA*-2xHA | This study |

**TABLE 2** Plasmids used in this study

| Plasmid | Relevant feature(s) | Reference |
|---|---|---|
| pWSK29 | Expression vector under the IPTG-induced lac promoter, AmpR[a] | Prof. dr hab. Dariusz Bartosik Institute of Microbiology, Department of Bacterial Genetics, University of Warsaw |
| pWSK29-sanA | pWSK29 vector with sanA sequence insert, AmpR | (11) |
| pFCcGi | Vector with arabinose-inducible expression of GFP and constitutive expression of mCherry | (35) |
| pFC-psicA | Based on pFCcGi; with constitutive mCherry expression and GFP expression under the control of sicA promoter | This study |
| pFPV25.1GFPmut3Kan | Plasmid with GFPmut3 under the constitutive rpsM promoter, multiple cloning site and hemagglutinin tag (HA), KanR | (36) |
| pFPV25.1GFPmut3Kan-sicA-2xHA | pFPV25.1GFPmut3Kan vector with sicA and its promoter insert, KanR | This study |
| p16Slux | Temperature-sensitive p16Slux plasmid (containing the luxABCDE operon of Photorhabdus luminescens), EryR | (37) |

[a]IPTG, isopropylthio-β-galactoside.

penicillin/streptomycin and seeded at a concentration of $1 \times 10^6$ or $2 \times 10^5$ cells per well in a 6-well or 24-well plate, respectively, 24 h before infection.

## Growth curves

To determine the growth curves, LB was inoculated with an individual bacterial colony and incubated overnight at 37°C with agitation at 180 rpm. The resulting cultures were then diluted to an $OD_{600}$ of 0.05 in LB and incubated until the early log-phase growth (EEP) ($OD_{600} = 0.5$) at 37°C, 220 rpm. Subsequently, cultures were centrifuged, rinsed, and resuspended in 0.9% NaCl. Optical density was assessed, and cultures were further diluted in LB to achieve a bacterial concentration of $5 \times 10^6$ CFU/mL. Optical density measurements were taken in Tecan Spark Control (Tecan) at 15 min intervals over 12 h, and the cultures were shaken 30 s before each measurement. The study was conducted with three independent biological replicates, and dilution series were set up on LB agar plates for verification of the initial bacteria number.

## Cloning of *sicA* and its promoter

All genes or their promoters were amplified from the *S*. Typhimurium 4/74 strain. Amplification was carried out by PCR using Phusion polymerase (Thermo) with primers

**TABLE 3** Primers used in this study

| Name | Sequence (5′–3′) | Reference |
|---|---|---|
| sanA-red-luc-for | CCGTTACGCCGGAACAATTGCTTGAACTGGAAAAGAAAAAAG GGAAATGAAGGAGGACAGCTATGGAAGACGCCAAAAACATAAGAA | This study |
| sanA-red-rev | AAGCGGGAGTAGCAGAAAGGCTAATATGACAAATATCGTCTGTACA TCCACGTGTAGGCTGGAGCTGCTTC | This study |
| sanA-check-seq-for | AGTGTTACGCGGTACCTTCAC | This study |
| sanA-check-seq-rev | CAATATTGTACGGGATCGGCAT | This study |
| 16S_rev_XhoI | CTGATCTCGAGGGCGGTGTGTACAAGG | (37) |
| 16S_fwd_int | ATTAGCTAGTAGGTGGGGTAACGGCTCACCTAGG | (37) |
| pFCcGi-sicA-for | ACATACGCGTGCGCCGCGTAAGGCAGTAGC | This study |
| pFCcGi-sicA-rev | ACATTCTAGATACTTACTCCTGTTATCTGTCACCG | This study |
| pFCcGi-seq-for | CATACTCCCGCCATTCAG | This study |
| pFCcGi-seq-rev | GTGTCTTGTAGTTCCCGTC | This study |
| sicA-2xHASac-for | ACAGAGCTCGCCGCGTAAGGCAGTAGC | This study |
| sicA-2xHABgl-rev | ACAGATCTTTCCTTTTCTTGTTCACTGTGCTG | This study |

listed in Table 3. PCR products were purified by the GeneJET PCR purification kit (Thermo), whereas plasmid DNA was isolated by the GeneJET Plasmid Miniprep Kit (Thermo). To create a dual reporter with constitutive mCherry expression and inducible GFP expression, the *sicA* promoter was inserted into the pFCcGi plasmid in MluI/XbaI digestion sites. For the HA-based reporter, *sicA* with promoter sequences was cloned into pFPV25.1GFPmut3.1Kan-2xHA in SacI/BglII digestion sites. The DNA sequence of all the inserts was confirmed by Sanger sequencing.

## Infection assay

Bacteria were grown under conditions optimizing SPI-1-dependent invasion or SPI-2-dependent replication within macrophages. For SPI-1-inducing conditions, an overnight culture was subcultured with aeration in LB at 37°C until the ESP (38). For SPI-2-inducing conditions, bacteria were grown until the LSP (overnight culture) with aeration (39). iBMDM monolayers were infected with stationary phase bacteria opsonized in mouse serum for 20 min using an MOI of 10:1. To synchronize the infection, the plates were centrifuged for 5 min at $165 \times g$, followed by a 30 min incubation at 37°C (5% $CO_2$). Fresh DMEM supplemented with 100 µg/mL of gentamicin (Gm) was added to kill extracellular bacteria, and the macrophage monolayers were incubated with added Gm for 90 min (40). After washing with DMEM, the monolayers were lysed in 1% Triton X-100 and diluted with phosphate buffered saline (PBS). Dilutions of the suspension were then plated on LB agar to assess the number of viable bacteria. To evaluate intracellular growth, the medium containing 100 µg/mL Gm was replaced with DMEM supplemented with 10 µg/mL of Gm, and parallel cell cultures were examined for viable bacteria 24 h following infection. Similarly, for the Caco-2 invasion assay, ESP bacteria were added to the monolayer until the final MOI = 100, cells were treated as described above, and plating dilutions of the suspension on LB agar determined CFU/mL of bacteria.

## Determination of SanA expression during infection

Transcriptional fusion was created as described previously by Gerlach et al. (41). Briefly, the p3121 plasmid, carrying luciferase, was used as a template and amplified with target gene-specific primers. PCR products were then purified, and the residual template plasmid was removed by a DpnI restriction digest. The resulting PCR product was analyzed by agarose gel electrophoresis and used for electroporation into competent cells of *S*. Typhimurium, harboring pKD46. Proper integration of the reporter cassette was confirmed by colony PCR using sanA-check-seq-for and sanA-check-seq-rev primers and Sanger sequencing. To determine SanA expression during infection, the *S*. Typhimurium *sanA*$_{RBS}$*::luc* strain proceeded in the infection assay with the use of iBMDM according to the protocol described above. After lysis with Triton X-100, an equal number of samples was used to determine CFU by dilutional plating, while the rest was collected by centrifugation for 3 min at $13,000 \times g$. The pellet was then resuspended in the lysis buffer (100 mM potassium phosphate buffer [pH 7.8], 2 mM EDTA, 1% [wt/vol] Triton X-100, 5 mg/mL bovine serum albumin, 1 mM dithiothreitol (DTT), 5 mg/mL lysozyme), incubated for 30 min on ice, and sonicated. Lysates were then analyzed by the addition of luciferase assay reagent (LAR) (Promega) in white microtiter plates using a Tecan plate reader and represented as relative light units per $2 \times 10^6$ bacteria. The luminescence was compared to the control, consisting of cells infected with *Salmonella* lacking the luciferase reporter.

## Hemagglutinin-based reporter gene assays and Western blotting

*S*. Typhimurium WT and Δ*sanA* strains were transformed with the pFPV25.1GFPmut3.1Kan-*sicA*-2xHA construct by electroporation according to Sambrook and Russell (42). For assays, transformants were grown O/N at 37°C, 180 rpm. The next day, cultures were diluted to an $OD_{600} = 0.05$ and grown in SPI-1-inducing conditions, as described above. At indicated time points, the equivalent of bacteria to $OD_{600} = 0.4$ was

collected by centrifugation for 4 min at 4°C, 16,100 × $g$ (43). For Western blot analysis, pellets were suspended in 100 µL of loading dye, incubated for 5 min at 95°C, and proceeded with SDS-PAGE in a 15% gel. The separated proteins were transferred using semi-dry transfer (Bio-Rad) onto nitrocellulose and blocked for 1 h at room temperature (RT) with 5% fat-free dry milk in PBST (PBS supplemented with 0.1% Tween-20). A 1:1,000 dilution of HA-Tag Rabbit mAb (Cell Signalling, C29F4) in PBST was used as a primary antibody, whereas the secondary antibody was anti-rabbit peroxidase diluted 1:5,000 in PBST (Sigma, A6154). The blots were developed with Clarity Western ECL Substrate (Bio-Rad). After the first blotting, the membrane was incubated with 30% $H_2O_2$ for 20 min at 37°C to inactivate peroxidase activity (44). Then, the membrane was washed two times with PBST and processed with Western blot as described above, however, with GFP Mouse mAb as a primary antibody diluted 1:1,000 in PBST (Cell Signalling, 4B10) and anti-mouse peroxidase (Dako, P0447) as a secondary antibody diluted 1:5,000. The Western blots were developed with the use of Chemidoc XRS+ and analyzed using Image Lab software (Bio-Rad).

### GFP-based reporter gene assays

The pFC-p$sicA$ plasmid, which allows for the detection of GFP under the control of the $sicA$ promoter and constitutive expression of mCherry, was constructed by cloning as described above. $S$. Typhimurium WT and Δ$sanA$ strains were transformed with constructs by electroporation according to the Sambrook and Russell protocol (42). For assays, transformants were grown O/N at 37°C, 180 rpm. The next day, cultures were diluted to an $OD_{600}$ = 0.05 and grown in SPI-1-inducing conditions as described above. At the indicated time points, $OD_{600}$ was measured, then equivalent to $3 \times 10^8$ bacteria was collected by centrifugation (6,000 × $g$, 5 min, room temperature), and washed with PBS. Next, bacteria were resuspended in PBS, fixed for 30 min in 4% paraformaldehyde (PFA) in the dark, and washed three times with PBS. Before analysis, bacteria were resuspended in PBS and filtered. Bacteria carrying pFCcGi empty plasmid (mCherry constitutive expression; GFP no expression) or empty plasmid induced by arabinose (mCherry constitutive expression; GFP induced expression) were used as negative and positive GFP controls, respectively. Cellular fluorescence was measured on the BD Fortessa II cell analyzer with Diva 8 software (Becton Dickinson, Franklin Lakes, NJ, USA) with a total of 10,000 events of the bacterial population (gated on forward scatter-H versus side scatter-H dot plots). GFP-positive cells were gated on mCherry-positive bacteria, and double-positive populations were further analyzed using FlowJo.

### Generation of an antibody against SanA

The peptide DHRFKHLYGLHRDHHHD, corresponding to amino acid residues 165–184 of SanA, was synthesized and coupled to keyhole limpet hemocyanin (KLH) by Davids Biotechnologie GmbH, Regensburg, Germany. The KLH-coupled peptide was used as the immunogen for the generation of rabbit antiserum, which was further purified according to the protocol used by the company. The optimal concentration of antibody determined for Western Blotting application was 20 µg/mL.

### Bacteria fractionation

Bacterial cultures were separated at the ESP into soluble and membrane fractions by a lysozyme-EDTA-osmotic shock, and furthermore, the inner and outer membranes were selectively extracted with Triton X-100 as described previously (45). Fractions were resuspended in 30% SDS with Laemmli buffer (50 mM Tris-HCl, pH 6.8, 2% SDS, 10% glycerol, 1% β-mercaptoethanol, 12.5 mM EDTA, 0.02% bromophenol blue) and proceeded with Western blotting. Briefly, samples were normalized according to the bicinchoninic acid assay (Thermo), separated by SDS-PAGE on a 12% gel, and transferred onto polyvinylidene fluoride (PVDF) membrane using a semi-dry transblot system (Bio-Rad). Next, the membranes were blocked for 1 h at room temperature with 5%

fat-free dry milk in PBST. They were then incubated overnight with a 1:100 dilution of OmpA Rabbit antiserum (as a marker of the outer membrane) and LepB Rabbit antiserum (as a marker of the inner membrane), both of which were gifts from Prof. R. Dalbey, Ohio State University (46). Additionally, a 1:500 dilution of SanA Rabbit polyclonal antibody in PBST was used. The blots were developed with Clarity Western ECL Substrate (Bio-Rad) with the use of Chemidoc XRS+ and analyzed using Image Lab software (Bio-Rad).

## *lux* marking of *S*. Typhimurium strains for *in vivo* imaging

To track the colonization of mice by *S*. Typhimurium, bioluminescent variants of the bacteria were created using the p16Slux system as described by Riedel et al. (37). This approach incorporates the temperature-sensitive p16Slux plasmid, which includes the *luxABCDE* gene cluster from *Photorhabdus luminescens*, into the bacterial chromosome at the 16S rRNA gene locus. Briefly, p16Slux was transformed into bacteria by electroporation using standard protocols. Integration of the plasmid was achieved by incubating positive clones at a temperature non-permissive to plasmid replication (42°C) under erythromycin-selective pressure. Colonies were then checked for light emission, and the integration of p16Slux was confirmed by PCR using primers 16S_rev_XhoI and 16S_fwd_int, yielding the expected 1,150 bp fragment.

## *In vivo* experiments

Male BALB/c mice (6–8 weeks old) from Anlab, Prague, Czech Republic, were infected with an appropriate concentration of bacterial suspension in 0.1 mL of PBS via intragastric administration. The investigation was divided into three parts to compare the WT and Δ*sanA* strains regarding (1) ability to colonize organs; (2) virulence level; (3) CI.

1. To determine the ability of the WT and Δ*sanA* strains to colonize organs, mice (10 mice per group) were infected with $10^7$ CFU/mouse of the respective strains and euthanized 5 days post-infection. Five mice from the control group received 0.1 mL of PBS and were treated following the same protocol as the experimental groups. Their spleens, livers, lymph glands, and small intestines were collected, homogenized in PBS, and the resulting homogenates were diluted in PBS and plated on LB+Strep for bacterial load determination.

2. Following infection ($10^7$ CFU/mouse) of WT::p16Slux and Δ*sanA*::p16Slux strains, groups of mice (seven mice per group) were observed every 24 h using Lago X—(Spectral Instruments Imaging, USA) and Aura Imaging Software v.4.0 to track bioluminescent signals. A control group received an inoculation of 0.1 mL of PBS. Mice were anesthetized with 5% isoflurane for whole-animal imaging, maintaining anesthesia throughout the process. The resulting bioluminescence images, representing light intensity through pseudocolors (red indicating the highest intensity and blue the lowest), were created by merging pseudo-color and gray-scale images using Aura Imaging Software v.4.0. Upon detecting bioluminescent signals, mice were immediately euthanized, regardless of signal strength. Previous research indicated that visible bacterial luminescence consistently correlated with symptoms of systemic disease, such as disheveled fur, inactivity, bent posture, lack of coordination, shaking, eye discharge, and mortality within 24 h (47). Mice without luminescent signals and showing no infection signs were also euthanized after 21 days once the final images were obtained. Their spleens, livers, lymph glands, and small intestines were collected and homogenized in PBS. The resulting homogenates were diluted in PBS and plated on LB+Strep for bacterial load determination.

3. The CI involved infecting mice with a mixed inoculum comprising both WT and Δ*sanA*::*kanR* in an equal ratio (48). For this, mice (10 mice per group, 5 for the control group) were treated with 20 mg of streptomycin, and after 24 h, a dose

equal to $10^7$ CFU/mouse was administered as described above. Homogenized tissue lysates from organs collected 5 days post-infection were simultaneously plated on LB+Strep supplied with or without kanamycin. In addition, feces were collected, homogenized, and plated as described above at 1 and 3 days post-infection. CI was calculated by dividing the number of Δ*sanA::kanR* CFU (kanamycin resistant) by the number of WT CFU (kanamycin sensitive).

## ACKNOWLEDGMENTS

This work was partially conducted using the equipment and the laboratories of the Regional Centre for Innovative Technologies in Food Production, Processing, and Safety.

The author(s) declare financial support was received for the research, authorship, and/or publication of this article. A.S. and K.G. were supported by the Polish National Science Centre Research Grant PRELUDIUM BIS number 2019/35/O/NZ6/01590. T.L.M.T. was funded by a Biotechnology and Biological Sciences Research Council David Phillips Fellowship BB/R011834/1 and Engineering and Physical Sciences Research Council (EPSRC) grant EP/X02377X/1, underwriting European Research Council Starting Grant.

## AUTHOR AFFILIATIONS

[1]Department of Biochemistry and Molecular Biology, Faculty of Veterinary Medicine, Wrocław University of Environmental and Life Sciences, Wrocław, Lower Silesian Voivodeship, Poland
[2]Faculty of Medicine, Wrocław Medical University, Wrocław, Lower Silesian Voivodeship, Poland
[3]Department of Infectious Disease, Centre for Bacterial Resistance Biology, Imperial College London, London, England, United Kingdom

## AUTHOR ORCIDs

Adrianna Stypułkowska http://orcid.org/0000-0002-5245-9206
Rafał Kolenda http://orcid.org/0000-0002-8145-579X
Ewa Carolak http://orcid.org/0000-0002-1762-7796
Joanna Czajkowska http://orcid.org/0000-0002-6750-409X
Agata Dutkiewicz http://orcid.org/0000-0001-5506-4114
Wiktoria Waszczuk http://orcid.org/0000-0002-5318-6107
Wiktoria Bińczyk http://orcid.org/0009-0004-6600-9259
Teresa L. M. Thurston http://orcid.org/0000-0001-6139-3723
Krzysztof Grzymajło http://orcid.org/0000-0002-1163-0679

## FUNDING

| Funder | Grant(s) | Author(s) |
| --- | --- | --- |
| Narodowe Centrum Nauki | 2019/35/O/NZ6/01590 | Adrianna Stypułkowska Krzysztof Grzymajło |
| Biotechnology and Biological Sciences Research Council | BB/R011834/1 | Teresa L. M. Thurston |
| Engineering and Physical Sciences Research Council | EP/X02377X/1 | Teresa L. M. Thurston |

## AUTHOR CONTRIBUTIONS

Adrianna Stypułkowska, Conceptualization, Investigation, Methodology, Validation, Visualization, Writing – original draft, Writing – review and editing | Rafał Kolenda, Conceptualization, Methodology, Writing – review and editing | Ewa Carolak,

Investigation | Joanna Czajkowska, Investigation | Agata Dutkiewicz, Investigation | Wiktoria Waszczuk, Investigation | Wiktoria Bińczyk, Investigation | Teresa L. M. Thurston, Resources, Writing – review and editing | Krzysztof Grzymajło, Conceptualization, Funding acquisition, Project administration, Resources, Supervision, Writing – original draft, Writing – review and editing

## DATA AVAILABILITY

No data were used for the research described in the article.

## ETHICS APPROVAL

The study followed the guidelines of the International Animal Care Convention and was approved by the Local Ethics Committee for Animal Experimentation in Wrocław, Poland (061/2022/P1 from 18/01/2023).

## ADDITIONAL FILES

The following material is available online.

### Supplemental Material

**Supplemental figure legends (Spectrum02833-24-s0001.docx).** All Supplementary Figures with captions.
**Figure S1 (Spectrum02833-24-s0002.tif).** Growth curves of *S. Typhimurium* 4/74 WT and sanARBS::luc in LB medium.
**Figure S2 (Spectrum02833-24-s0003.tif).** Densitometric analysis of protein bands imaged with the ChemiDoc MP.
**Figure S3 (Spectrum02833-24-s0004.tiff).** Fraction of cells expressing sicA. Fraction of cells in the ON state was determined relative to the negative control (100% in the OFF state), which consisted of the measured fluorescence of cells not expressing the GFP. EEP: early exponential growth phase corresponding to OD600=0.5; ESP: early stationary growth phase corresponding to OD600=2.0.
**Figure S4 (Spectrum02833-24-s0005.tiff).** Fraction of cells expressing sicA. Fraction of cells in the ON state was determined relative to the negative control (100% in the OFF state), which consisted of the measured fluorescence of cells not expressing the GFP.
**Figure S5 (Spectrum02833-24-s0006.tiff).** Monitoring of WT::p16Slux *S. Typhimurium* and ΔsanA::p16Slux infections in mice using whole-body bioluminescence imaging (BLI).

### Open Peer Review

**PEER REVIEW HISTORY (review-history.pdf).** An accounting of the reviewer comments and feedback.

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
