## [Reviewer comments · Microbiology Spectrum]

Microbiology Spectrum

SanA is an inner membrane protein mediating *Salmonella* Typhimurium infection

Adrianna Stypułkowska, Rafał Kolenda, Ewa Carolak, Joanna Czajkowska, Agata Dutkiewicz, Wiktoria Waszczuk, Wiktoria Bińczyk, Teresa Thurston, and Krzysztof Grzymajło

Corresponding Author(s): Krzysztof Grzymajło, Wrocław University of Environmental and Life Sciences

Review Timeline:

Submission Date:	November 4, 2024
Editorial Decision:	February 8, 2025
Revision Received:	March 3, 2025
Accepted:	March 4, 2025

Editor: Carlos Blondel

Reviewer(s): Disclosure of reviewer identity is with reference to reviewer comments included in decision letter(s). The following individuals involved in review of your submission have agreed to reveal their identity: Yosef Daniel Huberman (Reviewer #2)

Transaction Report:

DOI: <https://doi.org/10.1128/spectrum.02833-24>

Re: Spectrum02833-24 (SanA is an inner membrane protein mediating Salmonella Typhimurium infection)

Dear Dr. Krzysztof Grzymajlo:

Thank you for the privilege of reviewing your work. Below you will find my comments, instructions from the Spectrum editorial office, and the reviewer comments.

Please note that reviewer #1 provides a list of "Recommendations for Further Experiments". You don't need to perform such experiments, but it would be valuable to discuss the overall points made by the reviewer's proposal in the discussion section.

Revision Guidelines

Sincerely,
Carlos Blondel
Editor
Microbiology Spectrum

Reviewer #1 (Comments for the Author):

This manuscript further characterizes Salmonella typhimurium protein SanA, previously described as playing a role in modifying bacterial membranes. The authors investigate its localization, transcriptional expression patterns in vitro and in host cells, and the effects of sanA deletion on bacterial invasion and replication within host cells, as well as in vivo using a mouse infection model. While the study presents several interesting experimental findings (with well-designed assays), it does not provide

sufficient evidence to construct a cohesive understanding of SanA's role in Salmonella virulence mechanisms.

Summary of Observations

1. Localization and Function:

- o SanA is localized to the inner bacterial membrane.
- o Its deletion affects bacterial envelope permeability, antibiotic resistance, and intracellular replication.

2. In Vitro and In Vivo Fitness:

- o In vitro, the *sanA* mutant strain shows a fitness defect, as the wild-type strain out-competes it during co-culture.
- o In vivo, the *sanA* mutant is cleared more rapidly from BALB/c mice than the wild-type strain, indicating attenuated virulence.

3. Cellular Invasion and Replication:

- o During short-term infection, *sanA* deficiency enhances the invasion efficiency of host cells (epithelial and macrophage cell lines).

- o In the long term, the *sanA* mutant exhibits increased replication within host cells compared to the wild type.

4. Contradictions in Virulence Phenotype:

- o Despite reduced virulence and fitness in vivo, the *sanA* mutant shows increased CFU counts in lymph nodes and liver compared to the wild type.

5. Additional Observations:

- o The *sanA* mutant strain grows better in 0.5% yeast extract than in higher concentrations, though the relevance of this finding to virulence phenotypes remains unclear.
- o Increased *sicA* expression in the *sanA* mutant was observed, but no mechanistic explanation was provided to link this with the phenotype.

Major Concerns and Open Questions

1. Host Immunity and Intracellular Trafficking:

- o Does the *sanA* mutant exhibit increased susceptibility to host innate immune mechanisms?
- o Does it preferentially traffic to degradative compartments (e.g., lysosomes) or avoid elimination mechanisms within host cells?
- o Microscopic analysis of *S. typhimurium*-containing vacuoles (SeCVs) using classical trafficking markers would provide valuable insights.

2. Replication Mechanism:

- o How does the *sanA* mutant replicate more efficiently within host cells?
- o Is its division/replication mechanism normal, or does the mutation induce an alternative pathway?

3. Contradictory CFU Observations:

- o Why does the *sanA* mutant show higher CFU counts in lymph nodes and liver despite overall attenuated virulence and fitness defects?

- o Could this indicate the presence of a persistent or latent bacterial pool that is ultimately cleared by the immune system?

4. Envelope Stability and Nutrient Uptake:

- o The authors reinforce SanA's role in stabilizing the bacterial envelope, with its absence leading to increased drug permeability.
- o However, the improved growth of the *sanA* mutant in low (0.5%) but not higher yeast extract concentrations is intriguing but unexplained. The hypothesis of enhanced nutrient accessibility or uptake in the mutant remains speculative and warrants further investigation.

5. Regulatory Mechanisms and *sicA* Expression:

- o The increased *sicA* expression in the *sanA* mutant does not fully explain the observed phenotypes. Establishing a cause-and-effect relationship between *sicA* upregulation and the phenotypic changes would strengthen the study's conclusions.

Recommendations for Further Experiments

1. Characterize host-cell trafficking of the *sanA* mutant using microscopy and trafficking markers.
2. Investigate the mutant's replication mechanism and its impact on intracellular growth.
3. Explore the immune response in mice to determine if the increased CFU counts in lymph nodes and liver reflect a persistent bacterial population.
4. Examine the biochemical components of yeast extract to identify factors that influence mutant growth.
5. Investigate the regulatory pathways linking SanA to *sicA* expression and evaluate their mechanistic basis.

By addressing these gaps, the study could provide a clearer picture of SanA's physiological role and its contribution to Salmonella virulence mechanisms.

Reviewer #2 (Comments for the Author):

General comments:

In the present study, one *S. Typhimurium* strain was used together with some variants derived from the same WT strain. I suggest that authors limit the results and discussion to this serovar or even to this specific strain as other serovars or strains from the same serovars may behave differently, especially some host-specific serovars. Some assays were conducted with "at least three separate experiments". Authors should specify the minimum and maximum number of repetitions used and explain why it was not feasible to maintain a consistent number across all experiments.

Throughout the text, the same terms should be used: Lymph nodes or lymph glands.

Lines 130-136. The abbreviations LEP, ESP, LSP, and EEP should be explained in their first appearance in the manuscript.

Please revise

Line 170. Please change to "...analyzed strains was detected ...".

Line 171. Figure 2S. Please change "densitometric" to "densitometry".

Lines 249-251. Please change to "Bacterial membranes are composed of numerous proteins crucial in the interaction between the pathogen, the environment, and the host (16, 17). Among these molecules, SanA was first described....".

Line 283. Please change to "supporting an SPI-1-specific impact."

Line 295. Please omit (SCV) as it is not repeated in the text.

Lines 298-301. There is confusion regarding the host response to bacterial infection. If Δ sanA helps avoid lethal infection, it would be expected to find dead mice among WT-infected mice. According to the results, no mice died (only euthanized). Authors should explain this in coincidence.

Line 322. Please consider changing the word "promising" or adding a purpose/suggesting possible results to this claim.

Line 369. Please change to "...over 12 h...".

Line 493. Authors should specify in CFU the "appropriate" doses used and explain how these were determined/calculated.

Lined 497-501. The total number of mice is not clear. Were there two groups of 10 mice for WT and Δ sanA strains? What was the treatment for the control group? Figures 6 A-D describe different numbers of results (based on the colored dots, I find 7, 8, or 9). Please revise or explain the lack of coincidences between the description in the text and the figures. Authors should also explain why different numbers of mice were used in the trials.

Lines 718-728. Description Figure 2. Please change the order of the abbreviation for the same order of appearance in the figure. Please add the times of incubation for all four growth phases and Mg-MES.

Lines 510-518. The experimental design is not clear. As mice were euthanized upon observation of Salmonella, I suggest eliminating Figure 7A, which describes survival. This might be presented as "the first day with signs of infection and euthanization". I would also ask the authors to explain why it was decided to euthanize the mice and not continue with the imaging for a few more days. Important information regarding possible differences in colonization patterns and or time of bacterial persistence in the different tissues might have been demonstrated as well. UFC might be important but differences in concentration are also visible and compared qualitatively by the bioluminescence.

Table 1. I cannot locate the E. coli and SURE strains in the text. Please verify.

Figure 1S. Please omit non-relevant text: "Nc indicates medium without bacteria."

Figure 3S and 4S. Please add a description for the abbreviation GFP.

Figure 5S. Please change "sacrificed" to "euthanized". This figure is very similar to Figure 7 with more information. I suggest using this one instead of Figure 7.

The manuscript titled **“SanA is an inner membrane protein mediating Salmonella Typhimurium infection”** - regarding the role of the membrane protein SanA in *Salmonella* Typhimurium and its implications in the bacterium's pathogenesis, antibiotic resistance, and overall fitness. By exploring SanA's expression, localization, and its effects on invasiveness, the authors provide significant insights into a better understanding of bacterial membrane dynamics and interactions with host systems. The combination of *in vitro* and *in vivo* approaches is appreciated to support the hypothesis.

The manuscript is well-structured and clearly articulates the hypotheses, materials and methods, results, and interpretations. The writing is precise, although some sections may benefit from simplification for broader accessibility, particularly for readers less familiar with molecular techniques.

General comments:

In the present study, one *S. Typhimurium* strain was used together with some variants derived from the same WT strain. I suggest that authors limit the results and discussion to this serovar or even to this specific strain as other serovars or strains from the same serovars may behave differently, especially some host-specific serovars.

Some assays were conducted with “at least three separate experiments”. Authors should specify the minimum and maximum number of repetitions used and explain why it was not feasible to maintain a consistent number across all experiments.

Throughout the text, the same terms should be used: Lymph nodes or lymph glands.

Lines 130-136. The abbreviations LEP, ESP, LSP, and EEP should be explained in their first appearance in the manuscript. Please revise

Line 170. Please change to “...analyzed strains was detected ...”.

Line 171. Figure 2S. Please change “densitometric” to “densitometry”.

Lines 249-251. Please change to “Bacterial membranes are composed of numerous proteins crucial in the interaction between the pathogen, the environment, and the host (16, 17). Among these molecules, SanA was first described...”.

Line 283. Please change to “supporting an SPI-1-specific impact.”.

Line 295. Please omit (SCV) as it is not repeated in the text.

Lines 298-301. There is confusion regarding the host response to bacterial infection. If Δ sanA helps avoid lethal infection, it would be expected to find dead mice among WT-infected mice. According to the results, no mice died (only euthanized). Authors should explain this in coincidence.

Line 322. Please consider changing the word “promising” or adding a purpose/suggesting possible results to this claim.

Line 369. Please change to “...over 12 h...”.

Line 493. Authors should specify in CFU the “appropriate” doses used and explain how these were determined/calculated.

Lined 497-501. The total number of mice is not clear. Were there two groups of 10 mice for WT and $\Delta sanA$ strains? What was the treatment for the control group? Figures 6 A-D describe different numbers of results (based on the colored dots, I find 7, 8, or 9). Please revise or explain the lack of coincidences between the description in the text and the figures. Authors should also explain why different numbers of mice were used in the trials.

Lines 718-728. Description Figure 2. Please change the order of the abbreviation for the same order of appearance in the figure. Please add the times of incubation for all four growth phases and Mg-MES.

Lines 510-518. The experimental design is not clear. As mice were euthanized upon observation of Salmonella, I suggest eliminating Figure 7A, which describes survival. This might be presented as “the first day with signs of infection and euthanization”. I would also ask the authors to explain why it was decided to euthanize the mice and not continue with the imaging for a few more days. Important information regarding possible differences in colonization patterns and or time of bacterial persistence in the different tissues might have been demonstrated as well. UFC might be important but differences in concentration are also visible and compared qualitatively by the bioluminescence.

Table 1. I cannot locate the E. coli and SURE strains in the text. Please verify.

Figure 1S. Please omit non-relevant text: “Nc indicates medium without bacteria.”

Figure 3S and 4S. Please add a description for the abbreviation GFP.

Figure 5S. Please change “sacrificed” to “euthanized”. This figure is very similar to Figure 7 with more information. I suggest using this one instead of Figure 7.

Reviewer #1 (Comments for the Author):

This manuscript further characterizes *Salmonella typhimurium* protein SanA, previously described as playing a role in modifying bacterial membranes. The authors investigate its localization, transcriptional expression patterns in vitro and in host cells, and the effects of *sanA* deletion on bacterial invasion and replication within host cells, as well as in vivo using a mouse infection model. While the study presents several interesting experimental findings (with well-designed assays), it does not provide sufficient evidence to construct a cohesive understanding of SanA's role in *Salmonella* virulence mechanisms.

Summary of Observations

1. Localization and Function:

- o SanA is localized to the inner bacterial membrane.
- o Its deletion affects bacterial envelope permeability, antibiotic resistance, and intracellular replication.

2. In Vitro and In Vivo Fitness:

- o In vitro, the *sanA* mutant strain shows a fitness defect, as the wild-type strain out-competes it during co-culture.
- o In vivo, the *sanA* mutant is cleared more rapidly from BALB/c mice than the wild-type strain, indicating attenuated virulence.

3. Cellular Invasion and Replication:

- o During short-term infection, *sanA* deficiency enhances the invasion efficiency of host cells (epithelial and macrophage cell lines).
- o In the long term, the *sanA* mutant exhibits increased replication within host cells compared to the wild type.

4. Contradictions in Virulence Phenotype:

- o Despite reduced virulence and fitness in vivo, the *sanA* mutant shows increased CFU counts in lymph nodes and liver compared to the wild type.

5. Additional Observations:

- o The *sanA* mutant strain grows better in 0.5% yeast extract than in higher concentrations, though the relevance of this finding to virulence phenotypes remains unclear.
- o Increased *sicA* expression in the *sanA* mutant was observed, but no mechanistic explanation was provided to link this with the phenotype.

First of all, we would like to thank the reviewer for detailed revision of our work. We acknowledge the reviewer's concerns regarding the need for a more cohesive understanding of SanA's role in *Salmonella* virulence mechanisms. Our study aimed to characterize the contribution of SanA to bacterial membrane integrity, antibiotic resistance, intracellular replication, and virulence in vivo. While our findings present some seemingly contradictory phenotypes, we believe they contribute to a broader understanding of how *sanA* influences *Salmonella* pathogenesis. Below, we respond to the specific concerns highlighted in the review.

Major Concerns and Open Questions

1. Host Immunity and Intracellular Trafficking:

- o Does the *sanA* mutant exhibit increased susceptibility to host innate immune mechanisms?

Thank you for this insightful question. While we did not specifically assess this aspect in our experiments, we hypothesize that the *sanA* mutant may exhibit altered susceptibility to host innate immune mechanisms compared to the wild-type strain. Notably, changes in outer membrane composition can impact antibiotic sensitivity and drug resistance, underscoring its role in bacterial survival. These changes may also influence phagocytosis efficiency and intracellular survival within macrophages by increasing resistance to antimicrobial activities of host immune cells. This could explain the observed higher replication of the *sanA* deletion mutant within macrophages, as discussed in our previous work (DOI: [10.3389/fmicb.2023.1340143](https://doi.org/10.3389/fmicb.2023.1340143)).

We acknowledge the importance of further investigating the *sanA* mutant's interactions with host innate immune defenses. Future studies will incorporate assays such as antimicrobial peptide susceptibility tests and cytokine profiling (via ELISA or qRT-PCR, using *in vitro* and *in vivo* models) to better understand its susceptibility to immune responses and its role in *Salmonella* pathogenesis.

- o Does it preferentially traffic to degradative compartments (e.g., lysosomes) or avoid elimination mechanisms within host cells?

- o Microscopic analysis of *S. typhimurium*-containing vacuoles (SeCVs) using classical trafficking markers would provide valuable insights.

Thank you for this valuable insight. Indeed, microscopic analysis of *Salmonella*-containing vacuoles (SeCVs) using classical trafficking markers would provide critical insights into this aspect. Immunofluorescence microscopy with lysosomal markers such as LAMP-1 could help determine whether the *sanA* mutant is preferentially trafficked to lysosomes. Additionally, live-cell imaging using GFP-labeled *Salmonella* strains in combination with Rab5 and Rab7 markers would allow us to track the intracellular fate of the mutant bacteria in real time.

Other potential approaches include acidification assays with LysoTracker to assess vacuolar pH and protease protection assays to evaluate bacterial exposure to degradative compartments. However, it is important to note that this study was conducted as part of a doctoral thesis, and due to time and resource constraints, these advanced experiments were beyond the scope of this work. Nevertheless, we recognize their significance and consider them valuable directions for future research.

2. Replication Mechanism:

- o How does the *sanA* mutant replicate more efficiently within host cells?

The increased intracellular replication of the *sanA* mutant could be attributed to enhanced nutrient uptake, altered membrane permeability, or differences in host immune modulation and these aspects are discussed in lines 300-304 in the manuscript. We agree that additional experiments, such as metabolic profiling, bacterial division rate analysis, and intracellular localization studies, would be valuable in determining whether the mutation facilitates an alternative replication pathway or enhances bacterial fitness within host cells by modifying interactions with the intracellular environment.

However, it is important to emphasize that the primary objective of this study was to characterize the role of *sanA* in *Salmonella* virulence rather than to provide a detailed mechanistic explanation of its intracellular replication dynamics. Given that this work was conducted as part of a doctoral thesis, time

and resource constraints limited the scope of our investigations, preventing us from conducting these advanced experiments. Nonetheless, we acknowledge their importance and consider them promising directions for future research.

o Is its division/replication mechanism normal, or does the mutation induce an alternative pathway?

Thank you for your question. We did not evaluate this aspect in our study as it was beyond the intended scope. However, future research could employ live-cell imaging with fluorescent markers to track replication dynamics in real time, as well as single-cell sequencing or transcriptomics to identify potential pathway alterations. These approaches would provide deeper insights into whether the mutation facilitates an alternative replication mechanism or influences bacterial adaptation within host cells.

3. Contradictory CFU Observations:

o Why does the *sanA* mutant show higher CFU counts in lymph nodes and liver despite overall attenuated virulence and fitness defects?

Thank you for this valuable insight. We would like to highlight that in line with in vitro results, our in vivo data revealed that the Δ *sanA* strain exhibited increased colonization of the liver and lymph nodes, suggesting enhanced proliferation within the host. This phenotype aligns with our previous observations of heightened invasiveness linked to a 10-nucleotide *sanA* mutation, further supporting the connection between *sanA* expression and infection dynamics. Additionally, our findings suggest an association with SPI-1, a key virulence factor responsible for *Salmonella* invasion.

Moreover, despite the higher bacterial loads in organs, mice infected with the Δ *sanA* strain displayed increased survival rates compared to those infected with the wild-type strain. This supports the hypothesis that increased membrane permeability enhances immune recognition, triggering an immune response that limits bacterial dissemination without causing lethal infection. These information can be found in the Discussion section (303-306 lines) of the manuscript.

o Could this indicate the presence of a persistent or latent bacterial pool that is ultimately cleared by the immune system?

Thank you for your question. While our study did not specifically investigate the presence of a persistent or latent bacterial pool, this is an intriguing possibility. Addressing this question would require more advanced mechanistic studies, such as longitudinal infection models, single-cell analyses, or viability assays, to determine whether a subpopulation of bacteria persists before immune clearance. As this was beyond the scope of our current work, we propose it as a valuable direction for future research.

4. Envelope Stability and Nutrient Uptake:

o The authors reinforce *SanA*'s role in stabilizing the bacterial envelope, with its absence leading to increased drug permeability.

o However, the improved growth of the *sanA* mutant in low (0.5%) but not higher yeast extract concentrations is intriguing but unexplained. The hypothesis of enhanced nutrient accessibility or uptake in the mutant remains speculative and warrants further investigation.

Thank you for your comment. We address this aspect in the Discussion section (Lines 312-317), where we propose that the improved growth of the Δ *sanA* mutant in 0.5% yeast extract may result from increased membrane permeability, facilitating enhanced nutrient uptake and subsequent *sicA*

induction. In contrast, at higher nutrient concentrations (2% yeast extract), *sicA* expression levels were comparable between the mutant and wild-type strains, likely because nutrient availability was sufficient to support efficient transport in both backgrounds.

However, we recognize that further experiments are necessary to validate this hypothesis. Future studies could investigate whether specific metabolites are differentially accessible in the mutant, providing deeper insights into the metabolic consequences of *sanA* deletion.

5. Regulatory Mechanisms and *sicA* Expression:

o The increased *sicA* expression in the *sanA* mutant does not fully explain the observed phenotypes. Establishing a cause-and-effect relationship between *sicA* upregulation and the phenotypic changes would strengthen the study's conclusions

Thank you for your comment. To address our observations, we selected *sicA* as a specific marker and demonstrated that its upregulation correlates with the presence or absence of *sanA* at both the promoter and protein expression levels. However, we acknowledge that our study was not designed to establish a direct mechanistic link between *sicA* upregulation and the observed phenotypic changes.

Elucidating this relationship would require extensive mechanistic analyses, including functional complementation assays, targeted gene knockouts, and rescue experiments, to directly assess *sicA*'s role in driving these phenotypes. These experiments are technically complex, time-intensive, and beyond the intended scope of our current work. Nevertheless, we recognize their importance and suggest them as valuable directions for future research.

Recommendations for Further Experiments

1. Characterize host-cell trafficking of the *sanA* mutant using microscopy and trafficking markers.
2. Investigate the mutant's replication mechanism and its impact on intracellular growth.
3. Explore the immune response in mice to determine if the increased CFU counts in lymph nodes and liver reflect a persistent bacterial population.
4. Examine the biochemical components of yeast extract to identify factors that influence mutant growth.
5. Investigate the regulatory pathways linking *SanA* to *sicA* expression and evaluate their mechanistic basis.

By addressing these gaps, the study could provide a clearer picture of *SanA*'s physiological role and its contribution to *Salmonella* virulence mechanisms.

Thank you for this valuable suggestion. We agree that these experiments would greatly strengthen our study by offering deeper mechanistic insights. Recognizing their significance, we have incorporated these recommendations into the Discussion section (Lines 332-338) as future research directions, guiding readers toward potential avenues for further investigation.

Reviewer #2 (Comments for the Author):

General comments:

In the present study, one *S. Typhimurium* strain was used together with some variants derived from the same WT strain. I suggest that authors limit the results and discussion to this serovar or even to this specific strain as other serovars or strains from the same serovars may behave differently, especially some host-specific serovars.

Thank you for your suggestion. We acknowledge that different serovars and strains may exhibit variations in their behavior, particularly in host-specific contexts. However, based on our previous findings and bioinformatic analysis (DOI: [10.3389/fmicb.2023.1340143](https://doi.org/10.3389/fmicb.2023.1340143)), we have shown that *sanA* is widely conserved across diverse bacterial taxa, including Gram-negative and Gram-positive bacteria, with a notable presence in Gammaproteobacteria, Bacteroidetes, Actinobacteria, Clostridia, and Spirochaetia. Furthermore, our BLAST comparison revealed a 94% identity between SanA in *E. coli* and *Salmonella*, with 97% similarity in amino acid properties, indicating strong conservation. Given this high degree of similarity and its presence across multiple bacterial groups, we assume that *sanA* likely plays a similar role in all *Salmonella* strains.

Some assays were conducted with "at least three separate experiments". Authors should specify the minimum and maximum number of repetitions used and explain why it was not feasible to maintain a consistent number across all experiments.

Thank you for pointing this out. This was an oversight, and we confirm that all experiments were conducted in triplicate. We have revised the manuscript accordingly to ensure accuracy, consistency, and clarity.

Throughout the text, the same terms should be used: Lymph nodes or lymph glands.

We have updated the manuscript to ensure accuracy and consistency.

Lines 130-136. The abbreviations LEP, ESP, LSP, and EEP should be explained in their first appearance in the manuscript. Please revise

Thank you for your suggestion. We acknowledge this oversight and have now revised the manuscript to define LEP, ESP, LSP, and EEP at their first appearance for clarity.

Line 170. Please change to "...analyzed strains was detected ...".

The manuscript has been revised accordingly.

Line 171. Figure 2S. Please change "densitometric" to "densitometry".

The manuscript has been revised accordingly.

Lines 249-251. Please change to "Bacterial membranes are composed of numerous proteins crucial in the interaction between the pathogen, the environment, and the host (16, 17). Among these molecules, SanA was first described...".

The manuscript has been revised accordingly.

Line 283. Please change to "supporting an SPI-1-specific impact."

The manuscript has been revised accordingly.

Line 295. Please omit (SCV) as it is not repeated in the text.

The manuscript has been revised accordingly.

Lines 298-301. There is confusion regarding the host response to bacterial infection. If Δ sanA helps avoid lethal infection, it would be expected to find dead mice among WT-infected mice. According to the results, no mice died (only euthanized). Authors should explain this in coincidence.

Thank you for your comment. We acknowledge the potential confusion regarding the host response to Δ sanA infection. As discussed in the Discussion section (Lines 298-306), despite higher bacterial loads in organs, mice infected with the Δ sanA strain exhibited increased survival rates compared to those infected with the wild-type strain. This suggests that the heightened membrane permeability of Δ sanA enhances immune recognition, triggering a more effective immune response that limits bacterial dissemination without causing a fatal outcome.

Regarding the absence of mortality in WT-infected mice, we clarify that euthanasia was performed at predefined humane endpoints to prevent unnecessary suffering. This ethical approach ensured the humane treatment of animals while still allowing us to accurately assess infection progression and bacterial burden.

Line 322. Please consider changing the word "promising" or adding a purpose/suggesting possible results to this claim.

Thank you for your suggestion. We have removed the word "promising" to ensure clarity and precision in our statement.

Line 369. Please change to "...over 12 h...".

The manuscript has been revised accordingly.

Line 493. Authors should specify in CFU the "appropriate" doses used and explain how these were determined/calculated.

Thank you for your comment. The term "appropriate" was used to reflect the variation in CFU doses across different experiments, as detailed in 501-530 lines of the manuscript. Since each experiment required a specific bacterial load based on its design, we presented the CFU values accordingly in their respective sections rather than providing a single fixed number here.

Lined 497-501. The total number of mice is not clear. Were there two groups of 10 mice for WT and Δ sanA strains? What was the treatment for the control group? Figures 6 A-D describe different numbers of results (based on the colored dots, I find 7, 8, or 9). Please revise or explain the lack of coincidences between the description in the text and the figures. Authors should also explain why different numbers of mice were used in the trials.

Thank you for your comment. We confirm that there were 10 mice per strain (WT and *ΔsanA*, 20 total) plus 5 mice in the control group, which received 0.1 ml of PBS via intragastric administration. This information has now been incorporated into lines 503-504 for clarity.

The variation in the number of data points in Figures 6A-D is due to the removal of outliers, as described in the figure legend: Statistical differences were analyzed using the Mann-Whitney test after the removal of outliers (ROUT, Q=1%).

Regarding the different numbers of mice used in experiments, all trials were conducted in accordance with ethical committee guidelines and the 3R principle (approval number: 061/2022/P1), which emphasizes using the minimum number of animals required to obtain statistically significant results.

Lines 718-728. Description Figure 2. Please change the order of the abbreviation for the same order of appearance in the figure. Please add the times of incubation for all four growth phases and Mg-MES.

Thank you for your suggestion. We have changed the order of abbreviations to match their appearance in Figure 2 for consistency. However, instead of providing incubation times, we have chosen to present optical density (OD₆₀₀) values, as we consider this a more appropriate reference for readers. The time required to reach specific growth phases can vary between bacterial strains, making OD₆₀₀ a more standardized and comparable parameter.

Lines 510-518. The experimental design is not clear. As mice were euthanized upon observation of Salmonella, I suggest eliminating Figure 7A, which describes survival. This might be presented as "the first day with signs of infection and euthanization".

Thank you for your suggestion. We agree with your recommendation and have revised the description to "bioluminescence-free time" to more accurately reflect the experimental design.

I would also ask the authors to explain why it was decided to euthanize the mice and not continue with the imaging for a few more days. Important information regarding possible differences in colonization patterns and or time of bacterial persistence in the different tissues might have been demonstrated as well. UFC might be important but differences in concentration are also visible and compared qualitatively by the bioluminescence.

Thank you for this valuable insight. Based on our group's extensive experience in animal studies (over 10 years), we have previously confirmed that detectable bacterial luminescence consistently correlates with the onset of clinical signs of systemic disease. Our earlier studies demonstrated that once luminescence was detected, mice developed symptoms such as ruffled fur, lethargy, hunched posture, ataxia, tremor, eye discharge, and death within 24 hours, with higher signal intensities often associated with more rapid disease progression (<https://pubmed.ncbi.nlm.nih.gov/25914682/>).

Furthermore, our experimental design was conducted in accordance with bioethical committee guidelines, which approved our protocol to euthanize mice upon detection of luminescence to prevent unnecessary suffering (approval number: 061/2022/P1). While continued imaging might have provided additional insights into colonization patterns, our approach ensured compliance with ethical standards while still allowing for meaningful analysis through CFU quantification and bioluminescence assessment.

Table 1. I cannot locate the *E. coli* and SURE strains in the text. Please verify.

Thank you for your observation. We have removed the *E. coli* XL1-Blue and SURE strains from Table 1, as they were only used for construct generation, specifically for plasmid propagation and cloning purposes, and were not referenced in the main text.

Figure 1S. Please omit non-relevant text: "Nc indicates medium without bacteria."

Thank you for your suggestion. We have decided to retain this information as we consider it relevant for clarifying the treatment of negative controls in the experiment. This ensures transparency and consistency in data interpretation.

Figure 3S and 4S. Please add a description for the abbreviation GFP.

Thank you for your suggestion. We have added a description for the abbreviation "GFP" in the figure legends for Figures 3S and 4S to ensure clarity and consistency.

Figure 5S. Please change "sacrificed" to "euthanized". This figure is very similar to Figure 7 with more information. I suggest using this one instead of Figure 7.

Thank you for your suggestion. We have updated the manuscript, replacing "sacrificed" with "euthanized" for accuracy and ethical clarity.

Regarding Figure 7, we chose to keep it in the main text to present the most relevant information, while Figure 5S in the supplementary material provides additional details for readers who seek further context. This approach ensures clarity while maintaining accessibility to extended data.

Re: Spectrum02833-24R1 (SanA is an inner membrane protein mediating Salmonella Typhimurium infection)

Dear Dr. Krzysztof Grzymajlo:

Your manuscript has been accepted, and I am forwarding it to the ASM production staff for publication. Your paper will first be checked to make sure all elements meet the technical requirements. ASM staff will contact you if anything needs to be revised before copyediting and production can begin. Otherwise, you will be notified when your proofs are ready to be viewed.

Sincerely,
Carlos Blondel
Editor
Microbiology Spectrum